# CogniLoad: A Synthetic Natural Language Reasoning Benchmark With Tunable Length, Intrinsic Difficulty, and Distractor Density[*]

**Daniel Kaiser**[†‡] **Arnoldo Frigessi**[†§] **Ali Ramezani-Kebrya**[†§] **Benjamin Ricaud**[†‡]

[†]`m@dk.fo`

## Abstract

Current benchmarks for long-context reasoning in Large Language Models (LLMs) often blur critical factors like intrinsic task complexity, distractor interference, and task length. To enable more precise failure analysis, we introduce **CogniLoad**, a novel synthetic benchmark grounded in Cognitive Load Theory (CLT). CogniLoad generates natural-language logic puzzles with independently tunable parameters that reflect CLT's core dimensions: intrinsic difficulty ($d$) controls intrinsic load; distractor-to-signal ratio ($\rho$) regulates extraneous load; and task length ($N$) serves as an operational proxy for conditions demanding germane load. Evaluating 22 SotA reasoning LLMs, CogniLoad reveals distinct performance sensitivities, identifying task length as a dominant constraint and uncovering varied tolerances to intrinsic complexity and U-shaped responses to distractor ratios. By offering systematic, factorial control over these cognitive load dimensions, CogniLoad provides a reproducible, scalable, and diagnostically rich tool for dissecting LLM reasoning limitations and guiding future model development.

## 1 Introduction

Cognitive Load Theory (CLT) (Sweller, 1988) characterizes three types of cognitive load on human working memory when solving problems (Sweller, 1988; Paas et al., 2003; Lieder and Griffiths, 2020): intrinsic (ICL), extraneous (ECL), and germane (GCL). ICL stems from the inherent complexity and element interactivity of the task (Halford et al., 1998). ECL is induced by suboptimal task presentation requiring the processing of elements that are not task-relevant (Chandler and Sweller, 1991). GCL denotes the proportion of working-memory resources productively allocated to constructing and maintaining task-specific schemas (Paas et al., 2003; Sweller, 2010). Unlike ICL and ECL, which are properties of the task material, GCL reflects how the learner *allocates* resources; it must therefore be operationalized indirectly, through task conditions that demand sustained, constructive engagement with a task-relevant schema.

Large language models (LLMs) demand analogous computational resources when solving reasoning tasks. The essential element interactivity of a reasoning chain mirrors ICL, distractor elements reflect ECL, and sustained engagement with intrinsically relevant information over a long reasoning process acts as an operational proxy for GCL—the constructive effort to build, maintain, and repeatedly apply a coherent problem-state schema over many sequential steps.

To the best of our knowledge, no study has based the evaluation of problem-solving capacities of LLMs in CLT by distinguishing these three load types, and existing benchmarks often confound them: LongBench (Bai et al., 2024a) and L-Eval (An et al., 2024) vary context length but not necessarily the intrinsic reasoning depth; LogicBench (Parmar et al., 2024) probes ICL with minimal demands on ECL or context-induced load; BABILong (Kuratov et al., 2024) mixes multi-step reasoning with fixed distractor ratios, obscuring precise failure attribution.

---

[*]Project website: `https://cogniload.dk.fo`    [†]Integreat - Norwegian Centre for knowledge–driven machine learning    [‡]UiT - The Arctic University of Norway    [§]University of Oslo

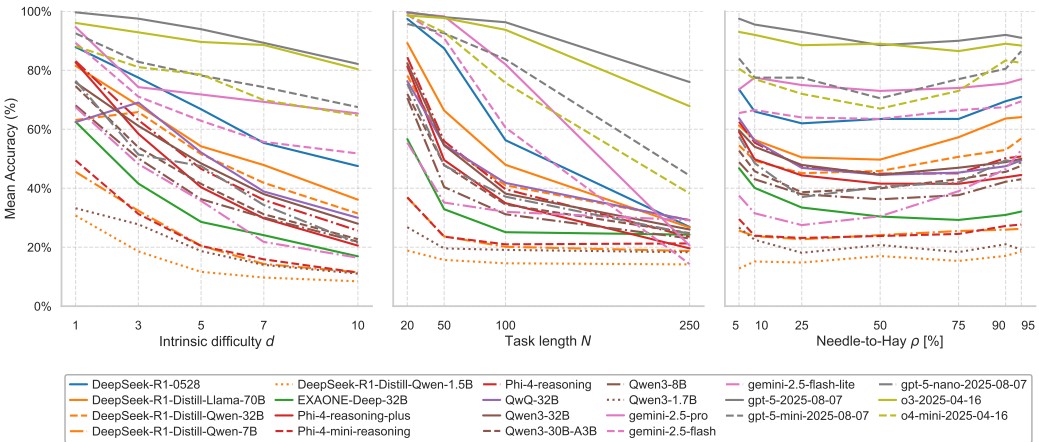

Figure 1: The average accuracy of models across the evaluated parameter space for $d \in \{1, 3, 5, 7, 10\}$ (left panel), $N \in \{20, 50, 100, 250\}$ (center panel), and $\rho \in \{5, ..., 95\}$ (right panel). Each plot selects one dimension for the X-axis and averages the accuracy of all evaluated puzzles for the other two dimensions relative to it.

We introduce **CogniLoad**, a controllable synthetic benchmark for long-context reasoning, inspired by CLT, that operationalizes these load types through tunable parameters in randomized natural-language logic puzzles: **(i) Intrinsic Load** via Intrinsic Difficulty $d$ controls the number of interacting entities, attributes, and logical clauses, directly manipulating ICL by varying essential element interactivity and reasoning depth. **(ii) Extraneous Load** via Distractor Density $\rho$ dictates distractor density; lower $\rho$ increases irrelevant elements, manipulating ECL. **(iii) Germane Load Proxy** via Task Length $N$: increasing the number of sequential statements scales the sustained schema maintenance required without changing per-step complexity or distractor density, serving as an operational proxy for conditions demanding germane-like cognitive work.

Our key contributions are summarized as follows:

1. We introduce *CogniLoad*, the first benchmark grounded in CLT that independently controls intrinsic load ($d$), extraneous load ($\rho$), and an operational proxy for germane load ($N$), scaling to arbitrarily long contexts.

2. We provide an algorithm for automatic randomized generation and evaluation of puzzle instances, enabling large-scale, reproducible, and contamination-resistant comparison of LLM capabilities.

3. We report empirical results on 22 state-of-the-art (SotA) reasoning LLMs (Figure 1), revealing distinct failure regimes across ($d, N, \rho$) and highlighting specific targets for model improvement.

Together, these contributions translate CLT into a precise diagnostic framework for understanding and advancing long-context reasoning in LLMs.

## 1.1 RELATED WORK

**Long-context Benchmarks (Working Memory Capacity).** A line of work starting with Long-Range Arena (LRA) (Tay et al., 2020) and followed by several recent benchmarks probe LLM performance on long sequences, often framed as testing "memory load" or context utilization. Earlier studies such as SCROLLS (Shaham et al., 2022), BookSum (Kryściński et al., 2021), and QMSum (Zhong et al., 2021) scale document length without manipulating intrinsic difficulty. LongBench (Bai et al., 2024a;b) and L-Eval (An et al., 2024) aggregate multi-task corpora up to 200k tokens, while BABILong (Kuratov et al., 2024), LongReason (Ling et al., 2025), RULER (Hsieh et al., 2024), ZeroSCROLLS (Shaham et al., 2023), and Michelangelo (Vodrahalli et al., 2024) increase context while the inherent difficulty of individual sub-tasks (ICL) may vary unsystematically and distractor

density (ECL) is often not a controlled variable. Consequently, performance degradation could be due to sheer length overwhelming processing capacity, or an inability to sustain germane-like cognitive work over extended relevant information, but the precise cause of failure is not clear.

**Logical-reasoning Benchmarks (Intrinsic Load).** A complementary line of benchmarks focuses on ICL by presenting tasks with high inherent complexity but often within minimal context lengths or distractors. Notable classical suites include ReClor (Yu et al., 2020), LogiQA (Liu et al., 2020), and BIG-Bench-Hard (BBH) (Suzgun et al., 2022). AutoLogic (Zhu et al., 2025) is a benchmark that explicitly focuses on scaling ICL through controllable complexity. LogicBench (Parmar et al., 2024), CLUTRR (Sinha et al., 2019), and ZebraLogic (Lin et al., 2025) also exemplify this by formulating symbolic logic puzzles that demand processing many interacting elements (e.g., multi-step deductions, handling negation, and constraint satisfaction). LogiEval (Liu et al., 2025) tests deductive, inductive, analogical, and abductive reasoning but forgoes independent dimensional control and synthetic generation. Similarly, mathematical reasoning datasets, e.g., GSM8K (Cobbe et al., 2021) and abstract rule induction tasks, e.g., ARC-AGI (Chollet et al., 2024) primarily escalate ICL by increasing the complexity of essential rules and their interdependencies.

**Needle in a Haystack Benchmarks (Extraneous Load).** Needle in a haystack (NIAH) designs (Gkamradt, 2023) specifically target ECL by embedding relevant facts ("needles") within large volumes of distractor text ("hay"). Variants such as Sequential NIAH (Yu et al., 2025) and Nolima (Modarressi et al., 2025) investigate the impact of such distractors, which constitute non-essential elements requiring processing for filtering, thereby imposing ECL. While these benchmarks effectively isolate the impact of distractors on information retrieval, the "needle" tasks themselves typically involve low ICL (e.g., simple fact lookup).

**Need for Multi-dimensional Evaluation.** CLT highlights the interplay of ICL, ECL, and germane processing under finite working memory (Paas et al., 2003). Existing LLM reasoning benchmarks, however, typically manipulate only one dimension without systematic, independent control over the others. Even benchmarks like MIR-Bench (Yan et al., 2025) which combine high ICL with extensive input, do not offer the factorial control needed to disentangle these loads, hindering precise diagnostics. Similar to our work, GSM-$\infty$ (Zhou et al., 2025) allows manipulating noise and difficulty. However, these parameters are not adjusted independently of task length.

## 2 BENCHMARK DESIGN: COGNILOAD LOGIC PUZZLES

### 2.1 PUZZLE DEFINITION AND CONSTRUCTION

CogniLoad is a family of natural-language logic-grid puzzles explicitly crafted to probe sequential reasoning capabilities of LLMs. The design goals are threefold: each puzzle (i) necessitates sequential multi-step deduction where order fundamentally matters; (ii) embeds a controllable number of relevant "needle" facts within the context of a controllable number of "hay" distractor statements; and (iii) provides parameters that control distinct dimensions of cognitive load. This section formalizes the task and describes the puzzle generation process, the control parameters, and key design choices.

Each puzzle in CogniLoad (see Figure 2) consists of a set of people with independent and mutable attributes. A series of statements, applied in strictly sequential order, updates these attributes according to conditions specified in each statement. The puzzle generation is parameterized by three key parameters: intrinsic difficulty $d$, total number of statements $N$, and needle-to-hay ratio $\rho$.

#### 2.1.1 BASIC PUZZLE CONSTRUCTION

A puzzle is formally characterized by the following components:

- **People**: A set $P = \{p_1, p_2, \ldots, p_n\}$ of persons in the puzzle, and $n = \max(d, 2)$.
- **Person of Interest (PoI)**: A randomly selected person $p^* \in P$ about whom the final question is asked.
- **Attribute Categories**: A set $A = \{c_1, c_2, \ldots, c_d\}$ of attributes randomly selected from a predefined taxonomy of 12 categories. Each category takes values in a Value Domain with a given finite cardinality, smaller or equal to 10.

---

**(i) Puzzle Instruction:** Solve this logic puzzle. You MUST finalize your response with a single sentence about the asked property (e.g., "Peter is in the livingroom.", "Peter is wearing blue socks",... ). Solve the puzzle by reasoning through the statements in a strictly sequential order.

**(ii) Initial State:**
- Brent is wearing green socks and is wearing purple gloves and last listened to classical music.
- Anthony is wearing purple socks and is wearing yellow gloves and last listened to disco music.
- ...

**(iii) Update Statements:**
1. The people wearing green socks listen to electronic music.
2. The people who last listened to classical music and wearing purple gloves put on yellow gloves.
3. ...

**(iv) Query:** What color of socks is Brent wearing?

---

Figure 2: Example CogniLoad puzzle with intrinsic difficulty $d = 3$, statements $N = 20$, and needle-to-hay ratio $\rho = 50\%$. Only a subset of the initial state and update statements is shown.

- **Value Domains**: For each category $c \in A$, a value domain $V_c = \{v_{c,1}, v_{c,2}, \ldots, v_{c,\ell_c}\}$ where $\ell_c = d + 1$ for $d > 1$ or $\ell_c = 3$ when $d = 1$. See Appendix G for the complete ontology.

- **State Function**: $S_t(p, c)$ represents the value of attribute $c$ for person $p$ at step $t$. Each person has values for the $d$ attribute of the selected attribute categories $A$, thus the state value represents a vector of dimension $d$.

### 2.1.2 PUZZLE INITIALIZATION

A puzzle starts with initialization statements ($t = 0$) that assign unique attribute values to each person: $\forall p \in P, \forall c \in A : S_0(p, c) \in V_c$ such that $\forall p_i, p_j \in P, i \neq j, \exists c \in A : S_0(p_i, c) \neq S_0(p_j, c)$. This initialization guarantees a maximally diverse starting point in which every pair of persons differs in at least one attribute, preventing trivial early states and ensuring that both needle and hay statements remain non-degenerate throughout generation.

### 2.1.3 STATEMENT GENERATION PROCESS

For each step $t$ from 1 to $N$, a statement is generated that changes the state of a person. If it updates the PoI, the statement is called a *needle*. An update for a non-PoI is called a *hay*.

1. **Statement Type Selection**: Given $N$ and $\rho$, let $n_{\text{needle}}^t$ and $n_{\text{hay}}^t$ be the remaining numbers of needles and hays to satisfy the desired proportion $\rho$ in the complete puzzle. The probability of selecting a needle statement is then $\mathbb{P}(T_t = \text{needle}) = n_{\text{needle}}^t / (N - t)$. The total number of needle statements in the puzzle is calculated as $n_{\text{needle}}^0 = \max(1, \min(N, \text{round}(N \cdot \rho/100)))$.

2. **Reference Person Selection**: Given the selected statement type $T_t$, the algorithm selects the reference person $r_t$: if $T_t = \text{needle} \implies r_t = p^*$ and if $T_t = \text{hay} \implies r_t \sim \text{Uniform}(P \setminus \{p^*\})$.

3. **Statement Structure**: CogniLoad samples a number of conditions $k_t \sim \text{Uniform}\{1, \ldots, d\}$ (the number of attribute categories that must simultaneously match for the update), a number of state updates $m_t \sim \text{Uniform}\{1, \ldots, d\}$ (the number of attribute categories modified when the rule fires), and uniformly samples attribute categories $C_t \subseteq A$, $|C_t| = k_t$ and state updates $U_t \subseteq A$, $|U_t| = m_t$.

4. **Condition and Update Value Specification**: For each category $c \in C_t$, the condition value is set by the reference person's current state: $v_{c,t} := S_{t-1}(r_t, c)$. For needles, these conditions target the PoI (and can possibly involve other people), while for hays the conditions can match anyone except the PoI. For update values, if $T_t = \text{needle} \implies u_{c,t} \sim \text{Uniform}(V_c)$ and if $T_t = \text{hay} \implies u_{c,t} \sim \text{Uniform}(V_c \setminus \{S_{t-1}(p^*, c)\})$.

5. **Logical Form**: The statement at step $t$ has the logical form:

$$\forall p \in P : \left( \bigwedge_{c \in C_t} S_{t-1}(p, c) = v_{c,t} \right) \Rightarrow \left( \bigwedge_{c \in U_t} S_t(p, c) = u_{c,t} \right).$$

Attributes not mentioned in the update set remain unchanged $\forall p \in P, \forall c \in A \setminus U_t : S_t(p, c) = S_{t-1}(p, c)$. This is not specified in the prompt but implicitly assumed by the LLMs.

Each statement is converted from its logical form into natural language via deterministic string templates (e.g., "The people wearing {color} socks listen to {genre} music."), ensuring full reproducibility without reliance on an LLM for text generation. The complete generation code is provided in the accompanying repository (see Section 5).

### 2.1.4 VALIDATION CONSTRAINTS

A sequence of validations verifies that the generated statement does not result in a state that prevents the generation of further needles and hays. If all validations pass, the statement is appended to the puzzle; otherwise a new statement is generated.

**For hay statements** $(r_t \neq p^*)$**:** After the update, the state of affected non-PoIs must not become identical to PoI $\forall p \in P \setminus \{p^*\}$ such that $\forall c \in C_t : S_{t-1}(p, c) = v_{c,t}, \exists c \in A : S_t(p, c) \neq S_t(p^*, c)$ and the update must not affect the PoI $\exists c \in C_t : S_{t-1}(p^*, c) \neq v_{c,t}$.

*Rationale.* Hay cannot affect the PoI and cannot collapse all non-PoIs onto the PoI: this ensures a hay statement is truly extraneous and that at least one non-PoI retains a different state from the PoI, permitting further hay generation.

**For needle statements** $(r_t = p^*)$**:** The update must not affect all non-PoI people $\exists p \in P \setminus \{p^*\} :$ $\exists c \in C_t : S_{t-1}(p, c) \neq v_{c,t}$ and after the update not all non-PoIs have identical states as the PoI $\exists p \in P \setminus \{p^*\} : \exists c \in A : S_t(p, c) \neq S_t(p^*, c)$.

*Rationale.* Needles cannot affect everyone and cannot collapse all non-PoIs to the PoI: this preserves state diversity and ensures that further genuine hay statements remain constructible.

To prevent the distractors from becoming too trivial to track at lower difficulties, we require that a hay statement does not result in all non-PoIs become identical so the set $P \setminus \{p^*\}$ must contain at least two persons with distinct attribute values. CogniLoad construction ensures that each hay statement $T_t = $ hay affects at least one non-PoI $\exists p \in P \setminus \{p^*\} : \forall c \in C_t : S_{t-1}(p, c) = v_{c,t}$.

### 2.1.5 FINAL QUESTION GENERATION

After all $N$ statements have been generated, the puzzle concludes with a question about a random attribute of the PoI, sampled as a random category $c_q \sim \text{Uniform}(A)$. The correct answer to the puzzle is $S_N(p^*, c_q)$ obtained from the final state of the PoI.

### 2.1.6 EVALUATION METRICS

We evaluate each puzzle by exact-matching of the queried attribute value in the model output, accommodating minor phrasing and common lexical variants (see Appendices D and E for the full pipeline). Specifically, we accept three outcome types as correct: (i) the gold value in the last sentence that names the PoI and contains a category qualifier; (ii) the gold value in the last sentence naming the PoI; and (iii) the gold value in the final sentence of the output. This graduated matching mitigates sensitivity to minor formatting drift while maintaining a deterministic, reproducible evaluation signal. The accuracy of model $M$ across the evaluation set is calculated as $\text{acc}(M) = \frac{1}{|Z|} \sum_{z \in Z} \mathbf{1}\big[\text{answer}_M(z) = S_N(p^*, c_q)\big]$ where $S_N(p^*, c_q)$ represents the final state value of the queried attribute $c_q$ for the PoI $p^*$ after all $N$ statements have been processed.

## 2.2 TUNABLE PARAMETERS

To systematically probe long-context reasoning, CogniLoad employs three independent parameters that operationalize CLT's load dimensions (Paas et al., 2003) and allow the creation of puzzles with varying characteristics. Together, they define the load profile of a puzzle instance.

**Intrinsic Difficulty** $(d)$    for $d \in \{1, 3, 5, 7, 10\}$ controls multiple facets of puzzle complexity (see Table 1), directly manipulating ICL which according to CLT hinges on element interactivity (Halford et al., 1998). Increasing $d$ increases ICL via: (i) combinatorial growth in state space ($\approx (d+1)^d$), (ii) increased interactivity between persons, attributes, and values, and (iii) increased rule complexity (up to $d$ conditions/updates per statement).

Table 1: Key parameters controlling the puzzle generation.

| Symbol | Name | Definition | Cognitive Load Affected |
|--------|------|------------|-------------------------|
| $d$ | Intrinsic Difficulty | Controls cardinality of people set $|P| = \max(d, 2)$, attribute categories $|A| = d$, for each category $c \in A$ the cardinality of value domains $|V_c| = \max(d + 1, 3)$, and the distribution of conditions and updates per statement: $k, m \sim \text{Uniform}\{1, ..., d\}$. | ICL: Element interactivity, state space/rule complexity. |
| $N$ | Task Length | Total number of sequential state transitions in the puzzle. | GCL Proxy / Task Length: Scales the number of sequential schema applications |
| $\rho$ | Needle-to-hay Ratio | Percentage of statements directly influencing the PoI (needles) versus distractor statements (hay) | ECL: Distractor density challenges filtering, selective attention, and imposing load from processing non-essential elements. |

**Task Length** ($N$) for $N \in \{20, 50, 100, 250\}$ sets the total number of sequential state-update statements. $N$ serves as an operational proxy for GCL: the schema to be maintained is the PoI's attribute vector across $d$ categories, and each statement requires a sequential update of this vector. Increasing $N$ scales the number of such updates *without* changing per-step element interactivity ($d$) or distractor density ($\rho$), thereby isolating the demand for sustained, constructive schema maintenance (Ericsson and Kintsch, 1995)—the hallmark of germane processing in CLT (Sweller, 2010).

**Needle-to-hay Ratio** ($\rho$) for $\rho \in \{5, ..., 95\}$ sets the percentage of PoI-relevant ("needle") versus distractor ("hay") statements, directly manipulating ECL. ECL arises from processing non-essential elements (Chandler and Sweller, 1991). Decreasing $\rho$ increases ECL via increased distractor density which challenges filtering. Increasing $\rho$ controls ECL by focusing resources on relevant information. Critically, CogniLoad's "hay" statements are syntactically similar to "needles" and involve valid state updates for non-PoIs, imposing a more challenging ECL than easy to distinguish distractor text.

## 3 RESULTS

We have evaluated the performance of 13 open weights LLMs on 100 random CogniLoad puzzles per $(d, N, \rho)$ configuration resulting in 14'000 puzzle instances per LLM in total. In addition, we have evaluated the proprietary Gemini-2.5 and gpt-5[1] models and DeepSeek-R1-0528 on 10 CogniLoad puzzles per configuration (i.e., 1'400 puzzle instances). The maximum context length (input + output) was set to 32K tokens and LLMs run with their preset default decoding settings and system prompts.

Figure 1 shows mean accuracy across models as each load dimension varies with trends corroborated by our regression analysis (Section 3.1). Error bars (90% Wilson confidence intervals) for all curves are provided in Appendix C.

**Intrinsic Difficulty** ($d$) Performance declines monotonically with $d$ for most models, although a few mid-tier models show small bumps at $d = 3$ (e.g., QwQ-32B: 0.62→0.69; DS-Qwen-32B: 0.63→0.66). Top models degrade only slightly from $d = 1$ to $d = 3$ (o3: 0.96→0.93; gpt-5: 1.00→0.97), while smaller or distilled models drop by 0.10-0.25 in the same range. By $d = 5$, 12 of 22 models fall below 50% accuracy. Beyond $d \geq 7$ the marginal decline flattens for the majority of models: the strongest models maintain their performance even at $d = 10$ (gpt-5: 0.82; o3: 0.80), and the weakest ones approach 0.10-0.15.

**Task Length** ($N$) This parameter remains the dominant stressor. Most models exhibit their steepest decline between $N = 20$ and $N = 50$ (e.g., DS-Llama-70B: 0.89→0.66; Qwen3-8B: 0.71→0.40),

---

[1] The gpt-5 family models were evaluated using the "medium" reasoning effort setting.

Table 2: Per-model quadratic-$\rho$ GLM estimates with Wald $z$ statistic for p-values alongside derived 50% load-capacity thresholds (see Section 3.1.3). The value $--$ for $NT_{50}$ indicates that no real root exists in $[0, 1]$. "DS" abbreviates "DeepSeek" in the model names. $***p<0.001$, $**p<0.01$, $*p<0.05$

| Model | $\beta_0$ | $\beta_d$ | $\beta_S$ | $\beta_\rho$ | $\beta_{\rho^2}$ | $ECL_{50}$ | $NT_{50}$ | $ID_{50}$ |
|---|---|---|---|---|---|---|---|---|
| gemini-2.5-pro | $22.51^{***}$ | $-0.41^{***}$ | $-9.15^{***}$ | $-1.67$ | $1.76$ | $153.3$ | $--$ | $12.74$ |
| gemini-2.5-flash | $18.14^{***}$ | $-0.44^{***}$ | $-7.56^{***}$ | $-1.79$ | $2.16$ | $111.5$ | $--$ | $8.56$ |
| gemini-2.5-flash-lite | $3.19^{***}$ | $-0.30^{***}$ | $-1.22^{***}$ | $-2.88^{**}$ | $3.82^{***}$ | $8.8$ | $0.93$ | $1.53$ |
| gpt-5-2025-08-07 | $17.34^{***}$ | $-0.39^{***}$ | $-5.11^{***}$ | $-7.04^{***}$ | $5.62^{***}$ | $382.8$ | $--$ | $14.78$ |
| gpt-5-mini-2025-08-07 | $11.09^{***}$ | $-0.22^{***}$ | $-3.96^{***}$ | $-4.87^{***}$ | $5.10^{***}$ | $164.1$ | $--$ | $11.72$ |
| gpt-5-nano-2025-08-07 | $6.50^{***}$ | $-0.31^{***}$ | $-2.43^{***}$ | $-4.30^{***}$ | $4.12^{***}$ | $35.7$ | $0.94$ | $2.87$ |
| o3-2025-04-16 | $12.83^{***}$ | $-0.22^{***}$ | $-4.26^{***}$ | $-2.72$ | $2.07$ | $356.9$ | $--$ | $19.07$ |
| o4-mini-2025-04-16 | $13.00^{***}$ | $-0.23^{***}$ | $-4.89^{***}$ | $-5.99^{***}$ | $6.37^{***}$ | $132.1$ | $--$ | $10.86$ |
| DS-R1-0528 | $13.70^{***}$ | $-0.39^{***}$ | $-5.28^{***}$ | $-4.13^{***}$ | $4.19^{***}$ | $104.6$ | $--$ | $7.51$ |
| DS-Llama-70B | $8.36^{***}$ | $-0.30^{***}$ | $-3.28^{***}$ | $-3.50^{***}$ | $3.92^{***}$ | $69.8$ | $0.53$ | $5.14$ |
| DS-Qwen-32B | $5.15^{***}$ | $-0.19^{***}$ | $-2.12^{***}$ | $-2.07^{***}$ | $2.29^{***}$ | $54.3$ | $0.78$ | $3.95$ |
| DS-Qwen-7B | $1.74^{***}$ | $-0.23^{***}$ | $-0.96^{***}$ | $-0.45$ | $0.58^{*}$ | $2.9$ | $--$ | $-0.53$ |
| DS-Qwen-1.5B | $-0.35^{**}$ | $-0.20^{***}$ | $-0.33^{***}$ | $0.47$ | $-0.14$ | $0.0$ | $--$ | $-3.95$ |
| Phi-4-reasoning-plus | $9.52^{***}$ | $-0.45^{***}$ | $-3.58^{***}$ | $-4.21^{***}$ | $3.41^{***}$ | $45.7$ | $0.16$ | $3.68$ |
| Phi-4-reasoning | $9.11^{***}$ | $-0.39^{***}$ | $-3.35^{***}$ | $-4.62^{***}$ | $3.99^{***}$ | $52.3$ | $0.92$ | $4.08$ |
| Phi-4-mini-reasoning | $1.70^{***}$ | $-0.24^{***}$ | $-0.81^{***}$ | $-1.40^{***}$ | $1.45^{***}$ | $1.3$ | $--$ | $-0.55$ |
| EXAONE-Deep-32B | $4.09^{***}$ | $-0.26^{***}$ | $-1.55^{***}$ | $-3.16^{***}$ | $2.45^{***}$ | $14.1$ | $--$ | $1.0$ |
| QwQ-32B | $5.68^{***}$ | $-0.21^{***}$ | $-2.08^{***}$ | $-3.70^{***}$ | $3.07^{***}$ | $48.0$ | $0.95$ | $3.56$ |
| Qwen3-32B | $7.22^{***}$ | $-0.29^{***}$ | $-2.75^{***}$ | $-3.21^{***}$ | $2.75^{***}$ | $53.9$ | $0.94$ | $4.1$ |
| Qwen3-30B-A3B | $5.83^{***}$ | $-0.30^{***}$ | $-2.25^{***}$ | $-2.96^{***}$ | $2.87^{***}$ | $36.7$ | $0.99$ | $3.05$ |
| Qwen3-8B | $5.40^{***}$ | $-0.26^{***}$ | $-2.19^{***}$ | $-2.87^{***}$ | $2.66^{***}$ | $30.8$ | $--$ | $2.2$ |
| Qwen3-1.7B | $0.62^{***}$ | $-0.17^{***}$ | $-0.46^{***}$ | $-1.53^{***}$ | $1.24^{***}$ | $0.0$ | $--$ | $-4.07$ |

while the best performing ones show relative resilience (gpt-5: 1.00→0.98; o3: 0.99→0.98). Accuracy declines with longer sequences as at $N = 100$, several models roughly halve their $N = 20$ accuracy (e.g., DS-Llama-70B: 0.48; Qwen3-32B: 0.38) and at $N = 250$ only two models show above 50% accuracy (i.e., gpt-5 at 0.76 and o3 at 0.68) while most perform at 0.20-0.30 accuracy.

**Extraneous Load / Needle-to-hay Ratio ($\rho$)** A characteristic U-shaped response emerges, with performance lowest around $\rho \in [25, 50]\%$ and recovering at higher $\rho$. Recovery equals or exceeds the low-$\rho$ baseline for several models (e.g., DS-Llama-70B: 0.61→0.64; gemini-2.5-flash-lite: 0.38→0.53). Frontier models show smooth, marginal variations (gpt-5: 0.97→0.89→0.91), while some models recover only partially (Phi-4-reasoning-plus: 0.59→0.45; EXAONE-Deep-32B: 0.47→0.32).

**Two competing effects underlie the U-shape.** Increasing $\rho$ simultaneously (a) reduces the number of distractor statements, making filtering easier, and (b) increases the number of PoI state transitions, making sequential tracking harder. The accuracy difference $\Delta_\rho = \text{Acc}(\text{low } \rho) - \text{Acc}(\text{high } \rho)$ reveals which effect dominates per model: $\Delta_\rho \approx 0$ (e.g., DeepSeek-R1, gpt-5-mini) indicates the two effects balance; $\Delta_\rho > 0$ (e.g., gpt-5, o3) indicates noise tolerance; $\Delta_\rho < 0$ (e.g., Gemini-2.5-flash, DS-R1-Llama) indicates distractor sensitivity outweighs the benefit of fewer essential steps. Appendix Appendix A further investigates this U-shape at different levels of $d$ and $N$, showing that increasing $d$ accentuates the U-shape while increasing $N$ shifts curves downward and progressively flattens them.

## 3.1 LOAD-SENSITIVITY REGRESSION

To quantify model-specific sensitivities of the accuracy to load dimensions and derive interpretable capacity thresholds for each model, we employ a regression-based approach that allows us to isolate the impact of each type of cognitive load (see Table 2).

### 3.1.1 REGRESSION MODEL SPECIFICATION

We model the performance of LLMs using a binomial generalized linear model (GLM) with a logit link function:

$$\Pr(Y{=}1) = \sigma\big(\beta_0 + \beta_d\, d + \beta_N\, \log_{10} N + \beta_\rho\, \rho + \beta_{\rho^2}\, \rho^2\big),$$

where the binary outcome $Y$ represents exact-match accuracy ($Y = 1$, when the model solves the puzzle correctly), $\sigma(\cdot)$ is the inverse logit function, and the coefficients $\beta_d$, $\beta_N$ and $\beta_\rho$ quantify sensitivity to intrinsic difficulty (ICL), task length (GCL), and distractor ratios (ECL), respectively. The inclusion of a quadratic term for $\rho$ is motivated by the characteristic U-shape observed in the third panel of Figure 1 and based on an improved Akaike Information Criterion (AIC) value for 18 out of the 22 fitted models when included (see Appendix F). Since $N$ ranges up to 250, we apply $\log_{10}$ to keep it at a similar scale as the other parameters of the regression.

### 3.1.2 Significance of Main Effects

In all models, $\beta_d$ and $\beta_N$ are significant and highly negative, confirming performance degradation with increased ICL and GCL. The quadratic term for $\rho$ is also significant (except for two models) confirming the U-shaped response for most models: models typically perform worst at intermediate $\rho$ values and recover as $\rho$ approaches either extreme. Five models exhibit statistically insignificant coefficients for $\rho$ terms, reflecting poor baseline performance for the smallest models and indifference to distraction for strong models (i.e., o3, gemini-2.5-pro, gemini-2.5-flash).

### 3.1.3 Capacity Points at 50% Accuracy

The GLM coefficients (Table 2) allow us to derive interpretable capacity thresholds. These represent the point at which a model's accuracy is predicted to drop to 50% when varying a single load parameter, while holding other load parameters at their estimated mean values:

**ECL$_{50}$** (Effective Context Length): Maximum number of statements a model can process while maintaining 50% accuracy. Large ECL$_{50}$ values indicate superior context handling.

**NT$_{50}$** (Needle-to-hay Threshold): Minimum proportion of relevant information required to maintain 50% accuracy. Crucially, *small* values indicate greater robustness to distractors. If the estimated NT$_{50}$ is missing, then the model accuracy is not expected to cross the 50% threshold for any value $0 \le \rho \le 1$, under mean conditions for $d$ and $N$.

**ID$_{50}$** (Intrinsic Difficulty): It is the maximum intrinsic complexity (number of interacting entities/attributes) that a model can handle while maintaining 50% accuracy. Negative values indicate failure to reach 50% accuracy even at the lowest difficulty setting under mean conditions for $N$ and $\rho$.

Mathematically, these thresholds are derived by setting the logit in the GLM equation to zero (for $\Pr(y = 1) = 0.5$) and solving for the parameter of interest, e.g.:

$$\text{ECL}_{50} = 10^{-(\beta_0 + \beta_d \bar{d} + \beta_\rho \bar{\rho} + \beta_{\rho^2} \bar{\rho}^2)/\beta_N}; \quad \text{ID}_{50} = -(\beta_0 + \beta_N \overline{\log_{10} N} + \beta_\rho \bar{\rho} + \beta_{\rho^2} \bar{\rho}^2)/\beta_d.$$

For NT$_{50}$, we solve the quadratic equation $\beta_0 + \beta_d \bar{d} + \beta_N \overline{\log_{10} N} + \beta_\rho \rho + \beta_{\rho^2} \rho^2 = 0$ for $\rho$.

### 3.1.4 Model Capacity

The regression analysis and estimated capacity thresholds (Table 2) reveal clear variations among models that can be grouped into three classes:

*Frontier/High-capacity Models:* gpt-5 and o3 lead by a wide margin (ECL50 > 300), followed by gemini-2.5-pro, gpt-5-mini, o4-mini, and DS-R1-0528. The high baseline performance of gemini-2.5-pro ($\beta_0 = 22.5$) together with the large $\beta_N$ of -9.15 is consistent with the uniquely large amount long context errors as $N$ increases (as illustrated in the Appendix E).

*Mid-capacity Models:* DS-Llama-70B , Qwen3-32B, DS-Qwen-32B, QwQ-32B, Phi-4-reasoning, Phi-4-reasoning-plus, Qwen3-30B-A3B, gpt-5-nano-2025-08-07, and Qwen3-8B form a broad middle tier with good performance at moderate $N$ and $d$ values.

*Low-capacity Models:* DS-Qwen-7B, Phi-4-mini-reasoning, DS-Qwen-1.5B, and Qwen3-1.7B exhibit minimal effective context handling capacity failing to reach 50% accuracy even under mean context/distractor conditions, deteriorating rapidly under slightly increasing load.

### 3.1.5 Differential Sensitivity to Load Dimensions

The estimated coefficients further reveal distinct sensitivity profiles:

*Sensitivity to context length* ($\beta_N$): Universally negative and highly significant, larger models often show greater relative degradation compared to their higher baselines. Yet large $\mathrm{ECL}_{50}$ values for frontier models arise from the combined effect of $\beta_0$, $\beta_N$ indicating comparisons are best made via $\mathrm{ECL}_{50}$ and not $\beta_N$ in isolation.

*Sensitivity to intrinsic difficulty* ($\beta_d$): Negative across models with a narrow range, it suggests a more uniform effect. Despite some steep $\beta_d$ values, high baselines (e.g., gemini-2.5-flash) yield large $\mathrm{ID}_{50}$ values unlike smaller models with similar $\beta_d$ (e.g., Phi-4-reasoning-plus).

*Sensitivity to information relevance* ($\beta_\rho$ and $\beta_{\rho^2}$): Confirms the U-shaped response, but $\mathrm{NT}_{50}$ values reveal nuanced distractor robustness variations masked by aggregate scores (e.g., DS-Llama-70B vs. Qwen3-32B). For frontier models, the absence of $\mathrm{NT}_{50}$ indicates achieving above 50% accuracy while for weak models the same absence indicates remaining below 50% accuracy.

### 3.1.6 INTERACTION EFFECTS AMONG LOAD DIMENSIONS

To probe whether load dimensions compound, we fit per-model logistic GLMs augmented with all pairwise and three-way interactions among $d$, $\log_{10} N$, and $\rho$ (full specification and coefficient table in Appendix B). Three robust patterns emerge:

*Super-additive difficulty–length coupling.* A negative $d \times N$ interaction is significant ($p_{\mathrm{LR}} < 0.01$) in 17/22 models, indicating that longer, more complex tasks degrade accuracy beyond what either factor alone would predict. Frontier models (gpt-5, o3) show no detectable $d \times N$ interaction, consistent with near-separable responses at current loads.

*Difficulty amplifies distractor harm.* A negative $d \times \rho$ is significant in 13/22 models: higher intrinsic complexity makes models more vulnerable to extraneous load, consistent with more complex schemas being more fragile under noise.

*Length–distractor trade-off is model-dependent.* The $N \times \rho$ interaction is significant in a subset of models and its sign separates two regimes: positive values (e.g., Qwen3-30B-A3B, EXAONE-Deep-32B) indicate that reducing distractors mitigates the length penalty, while negative values (e.g., DS-Qwen-7B) indicate that longer sequences amplify distractor harm. A positive three-way $d \times N \times \rho$ in 12/22 models shows that increasing the needle share is most beneficial when tasks are simultaneously long and complex.

## 3.2 FAILURE MODES ACROSS MODELS, LENGTH, AND DIFFICULTY

We analyze error categories from the evaluation pipeline (see Appendix D) to identify failure modes and provide the complete distributions and per-model breakdowns in the Appendix E.

**State-tracking mistakes dominate under load.** Across models, the most common non-context failure is wrong final attribution in the last valid PoI sentence (valid-logic), consistent with mis-tracking sequential updates rather than formatting issues. For example, at $N$=250, Qwen3-32B has 2'541 valid-logic cases, DS-Llama-70B 2'465, and QwQ-32B 2'092. These logic errors also increase monotonically with $d$ for nearly all models.

**Long-context budget overflows are a model-specific failure at extreme $N$.** The max-context errors grow sharply with $N$ for some models: gemini-2.5-flash (280 errors in 350 samples at $N = 250$) or gemini-2.5-pro (268/350). OpenAI models also make these errors at $N = 250$ but at much lower levels (gpt-5: 32/350; o3: 24/348). Token-length distributions (Appendix Appendix H) show that these overflows stem from Gemini's substantially higher output verbosity rather than the puzzles not generally being impossible to be solved within the provided 32K context limit. Because overflows are concentrated in specific model–condition pairs (primarily Gemini 2.5 at $N = 250$) while the vast majority of models and conditions remain well below the 32K budget, we report them as a separate error category. Gemini's accuracy at $N = 250$ should accordingly be interpreted as a lower bound on reasoning capability. For frontier models such as gpt-5 and o3, total token usage sits comfortably below the budget at the same $(N, d)$ conditions, confirming that their degradation reflects reasoning under load rather than serving limits.

**Answer-formatting drift emerges under higher $N$ and $d$, mainly in smaller models.** While poi-logic stays near zero for most models, last-logic increases notably for compact models (e.g., Phi-4-mini-reasoning: 400 last-logic at $N = 250$; DS-Qwen-7B: 116), indicating that under load,

models fail to state their answer in the required format (a single sentence naming the PoI and the queried attribute), even when they reason about the PoI elsewhere in their output.

**"Other" failures rise with sequence length in small and mid-tier models.** At $N = 250$, DS-Qwen-1.5B has 605 "other" cases (often claiming the puzzle is unsolvable) and DS-Qwen-7B 461 indicating a shift from precise (but wrong) answers to non-answers as load grows.

# 4  DISCUSSION

By operationalizing CLT, CogniLoad enables multi-dimensional LLM evaluation that reveals nuanced failure patterns obscured by single-dimensional benchmarks. Task length ($N$) emerges as the dominant constraint; models exhibit distinct sensitivities to $d$ versus $\rho$, with characteristic U-shaped responses to intermediate distractor densities; and estimated capacity thresholds provide concise "cognitive fingerprints" for diagnostic LLM evaluation. We discuss limitations below.

**Nuances of the CLT-LLM Analogy** While CLT provides a powerful analogous framework, it is crucial to acknowledge that "cognitive load" in LLMs manifests as computational constraints (e.g., attention saturation, representational bottlenecks) rather than biological working memory limitations. Our operationalization of $N$ as a proxy for conditions demanding GCL, for example, is an abstraction. Future research should aim to bridge CLT concepts with direct, mechanistic measures of the underlying computational processes in LLMs to refine this analogy.

**Scope of Reasoning and Generalizability** CogniLoad focuses on sequential and pure deductive reasoning without requiring domain knowledge. While this reasoning type is fundamental to various subject areas (e.g., code, math), it is distinct from alternative reasoning paradigms like inductive, abductive, or analogical reasoning. Extending the CLT-grounded multi-dimensional evaluation to other reasoning types and evaluating it in other languages is a promising next step.

**Beyond Accuracy and Main Effects** The current evaluation relies on exact-match accuracy. Future iterations could incorporate richer metrics (e.g., step-wise reasoning fidelity, solution coherence, uncertainty of solutions). Reinforcement learning on verifiable rewards (Guo et al., 2025) presents a promising application of CogniLoad in LLM training, as its generated metadata enables precise verification of reasoning steps despite limited data of this kind.

Despite these considerations, by decomposing the "task difficulty" into principled, controllable dimensions inspired by CLT, CogniLoad provides a more insightful perspective than single-dimensional benchmarks. It allows a more differentiated understanding of LLM reasoning capabilities and limitations, paving the way for more targeted development of robust and generalizable AI systems.

# 5  CONCLUSION

We introduced **CogniLoad**, a novel synthetic benchmark grounded in CLT for multi-dimensional evaluation of LLM long-context reasoning. By independently controlling parameters for intrinsic cognitive load ($d$), extraneous cognitive load ($\rho$), and task length ($N$), CogniLoad offers unprecedented diagnostic precision. Our evaluations revealed task length as a dominant performance constraint and uncovered unique "cognitive fingerprints" of LLM sensitivities to different load types, providing actionable insights beyond single-dimensional benchmarks. CogniLoad offers a reproducible, scalable, and theoretically-grounded tool to systematically dissect LLM reasoning limitations and guide the development of more capable and robust AI systems. While human and artificial cognition are mechanistically distinct, applying frameworks such as CLT to AI evaluation can provide valuable perspectives for understanding and characterizing their operational differences and capabilities.

## AUTHOR CONTRIBUTIONS

D.K conceived the idea, designed and implemented the generation algorithm, prepared the dataset, scored the LLMs, analysed the empirical results, designed and implemented the evaluation pipeline, and wrote the first draft of the manuscript. B.R. compiled the first draft of the literature overview and acquired the computational resources for the evaluation. All authors contributed refining the benchmark design and in editing the manuscript towards the final version.

## LLM DISCLOSURE

LLMs were used only in the early stages of writing the paper to refine phrasing and correct grammar. During coding they were used to update the chart formatting code, and to translate the manual implementation of the generation algorithm in Elixir to python for reproducibility.

## REPRODUCIBILITY

The project website is available at: `https://cogniload.dk.fo`.

The code for generating CogniLoad puzzles according to the algorithm described in this paper is provided at: `https://github.com/kaiserdan/cogniload`. The dataset of the puzzles on which the LLMs were evaluated for the results presented in this paper is provided on HuggingFace: `https://huggingface.co/datasets/cogniloadteam/cogniload`

## ACKNOWLEDGEMENTS AND SUPPORT

This work was supported by the Research Council of Norway through its Centres of Excellence scheme, Integreat - Norwegian Centre for knowledge-driven machine learning, project number 332645.

Ali Ramezani-Kebrya was supported by the Research Council of Norway through FRIPRO Grant under project number 356103.

We acknowledge NRIS Norway for awarding this project access to the LUMI supercomputer, owned by the EuroHPC Joint Undertaking, hosted by CSC (Finland) and the LUMI consortium through Sigma2, project number nn12027k.

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

# APPENDIX

The supplementary material is organized as follows:

- Appendix A investigates $\rho$ relative to $N$ and $d$.
- Appendix B analyses and discusses interaction effects among $d$, $N$, and $\rho$
- Appendix C charts model performance reported with error-bars.
- Appendix D discusses the evaluation, scoring pipeline, and error taxonomy.
- Appendix E reports the distribution of error types across models and different levels of $d$ and $N$.
- Appendix F compares the AIC values of linear vs quadratic $\rho$ in the GLM regression.
- Appendix G details the complete attribute ontology.
- Appendix H lists the token-length distribution of prompts and output models per model by $d$ and $N$.

## A  INVESTIGATING $\rho$ RELATIVE TO $N$ AND $d$

In Figure 1 of the paper we discover a characteristic U-shape of the performance of LLMs on CogniLoad relative to $\rho$ (i.e., the ratio of distractors to essential elements). The following figures investigate this U-shape at different levels of difficulty (Figure 3) and statement length (Figure 4).

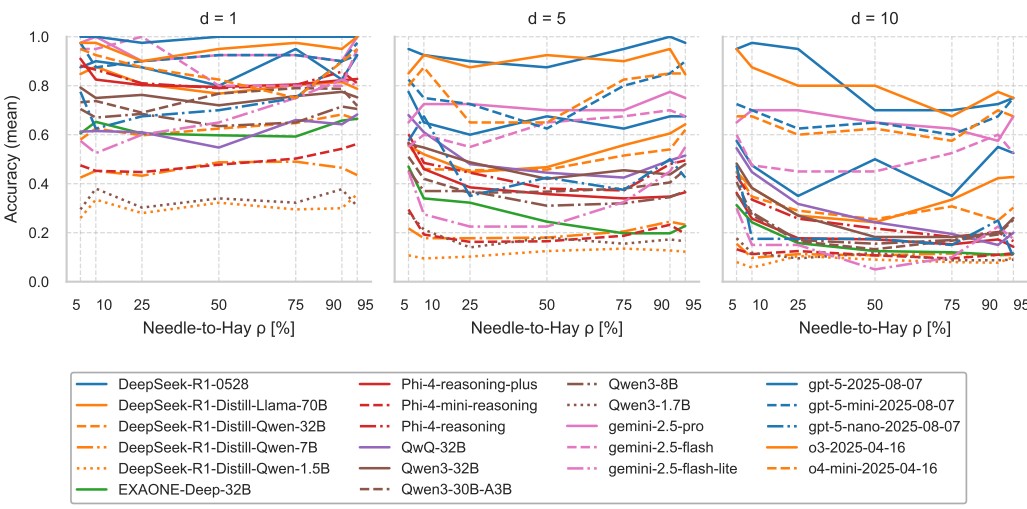

Figure 3: Distractor ratio $\rho$ relative to $d$ for $d \in \{1, 5, 10\}$

Figure 3 plots mean accuracy as a function of the needle-to-hay ratio $\rho$ (higher $\rho$ means more signal per distractor) under three difficulty settings $d = 1, 5, 10$. For the easiest tasks ($d = 1$, left) accuracies are high and largely flat; most models show at most a shallow U-shape, with a mild dip around $\rho = 25\,\%$–$50\,\%$ and a small recovery by $\rho \geq 75\,\%$. At intermediate difficulty ($d = 5$, centre) the trough widens: many curves drop sharply from $\rho = 10\,\%$ to $25\,\%$ and bottom out across $\rho \approx 25\,\%$–$50\,\%$ before rebounding at high $\rho$. A notable exception is EXAONE-Deep-32B, which trends downward as $\rho$ increases for $d \geq 5$. Under the hardest condition ($d = 10$, right) capability tiers separate. Frontier models remain robust and still recover at high $\rho$—averaged over $\rho$, GPT-5 $0,821$, O3 $0,804$, GPT-5-mini $0,675$, O4-mini $0,646$, and Gemini-2.5-Pro $0,654$—whereas mid-size and small models stay low despite a late uptick (e.g., DeepSeek-R1-0528 $0,475$; DeepSeek-R1-Distill-Llama-70B $0,361$; QwQ-32B $0,300$; Qwen3-32B $0,280$; Phi-4-reasoning $0,256$; Qwen3-1,7B $0,110$). Overall, the panels show that (i) accuracy shifts downward and the U-shape becomes more pronounced as $d$ increases; (ii) recovery at high $\rho$ is strongly model-dependent and favours larger, reasoning-tuned

systems; and (iii) the moderate-$\rho$ regime ($\rho \approx 25\,\% - 50\,\%$) remains the most adversarial, where balancing signal and noise is hardest.

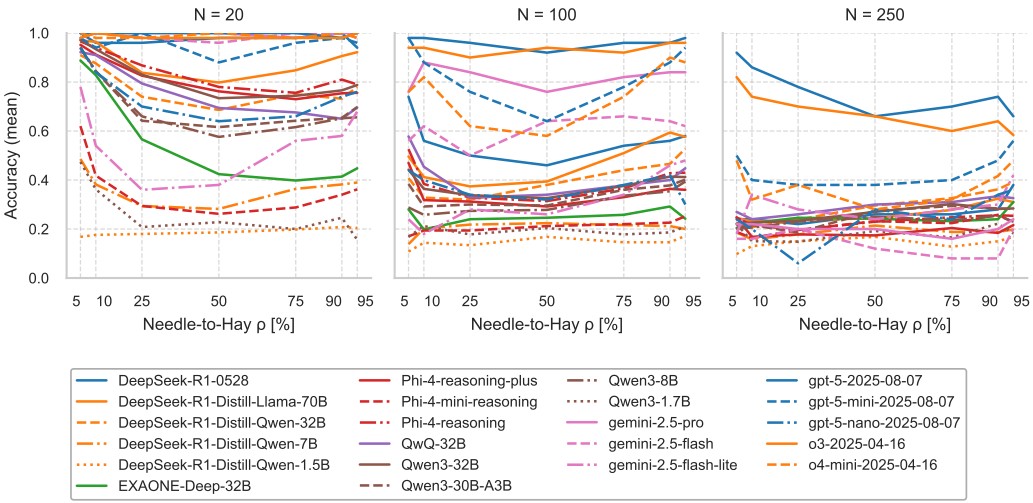

Figure 4: Distractor ratio $\rho$ relative to $N$ for $N \in \{20, 100, 250\}$

Figure 4 shows mean accuracy as a function of the needle-to-hay ratio $\rho$ for $N = 20$ (left), $N = 100$ (centre), and $N = 250$ (right). At $N = 20$ many systems still display a clear U-shape: accuracy begins near $1, 0$, dips at medium clutter around $\rho \approx 25 - 50\,\%$, and then rebounds as needles dominate. As the context grows, this dependency compresses and then largely disappears. At $N = 100$ the strongest models are nearly $\rho$-invariant (e.g., curves remain between about $0, 93$ and $0, 97$), while mid-tier models retain only a shallow trough. By $N = 250$ most curves are flat or gently increasing with $\rho$ at much lower levels; only the frontier models exhibit a modest late uptick.

Taken together, the two figures show that difficulty $d$ and length $N$ both depress accuracy but modulate its dependence on $\rho$ in different ways. Increasing $d$ accentuates the classic U-shape (i.e., many models drop from $\rho = 10\,\%$ into the mid-range $\rho \approx 25\,\% - 50\,\%$ and then recover as needles dominate) whereas increasing $N$ shifts the whole curve downward and progressively flattens it. These stresses reveal a clear stratification: frontier models remain comparatively robust (retaining some high-$\rho$ gains and partial $\rho$-invariance), a mid-tier exhibits a pronounced but narrowing mid-$\rho$ trough as $N$ grows, and small models settle at low, nearly $\rho$-insensitive performance. In short, the mid-$\rho$ band $25 - 50\,\%$ is the most adversarial at fixed $N$ and $d$, but at long contexts the primary failure mode becomes $N$ itself rather than $\rho$.

## B  INTERACTION EFFECTS AMONG $d$, $N$, AND $\rho$

To probe whether load dimensions compound each other in predictable ways, we fit, for each model, a logistic GLM that augments the main-effects specification with all pairwise and three-way interactions among intrinsic difficulty $d$, effective length $\log_{10} N$, and the needle ratio $\rho$, while retaining a quadratic main effect for $\rho$ to capture the observed U-shape:

$$\text{logit } P(\text{correct}) \; = \; \beta_0 + [\, d, \log_{10} N, \rho \,] + [\, d{:}\log_{10} N, \, d{:}\rho, \, \log_{10} N{:}\rho \,] + [\, d{:}\log_{10} N{:}\rho \,] + \rho^2.$$

In Table 3 we report for each interaction the coefficient $\beta$ and an ANOVA-style likelihood–ratio $p$-value ($p_{\text{LR}}$) obtained by dropping that interaction from the full model.

### B.1  GLOBAL PATTERNS

Three robust patterns emerge across models:

- A strong $d \times N$ interaction is widespread and negative: 17/22 models show $p_{\text{LR}} < 0.01$ with $\beta_{d \times N} < 0$ (e.g., QwQ-32B: $-0.39$, $p_{\text{LR}} = 1.25\mathrm{e}{-33}$; Phi-4-reasoning: $-0.45$,

$p_{\mathrm{LR}} = 1.94\mathrm{e}{-}30$). This indicates that increases in intrinsic difficulty and sequence length compound to reduce accuracy additively. Frontier systems (gpt-5, o3) show no detectable $d \times N$ interaction (all $p_{\mathrm{LR}} > 0.3$), consistent with their near-separable responses to $d$ and $N$ at current loads.

- A clear $d \times \rho$ interaction is present in 13/22 models (all $\beta_{d \times \rho} < 0$, $p_{\mathrm{LR}} \leq 0.01$ in those cases; e.g., QwQ-32B: $-0.96$, EXAONE-Deep-32B: $-0.89$). This confirms that higher intrinsic difficulty exacerbates the harm of distractors. Several strong models remain $\rho$-invariant (e.g., gpt-5, o3; $p_{\mathrm{LR}} > 0.2$), and DS-Llama-70B is marginal ($p_{\mathrm{LR}} = 0.055$).

- The $N \times \rho$ interaction is model-dependent and less ubiquitous (significant in 4/22). Two qualitatively distinct regimes appear: for some mid-tier models the coefficient is positive and significant (Qwen3-30B-A3B: $+1.16$, $p_{\mathrm{LR}} = 7.18\mathrm{e}{-}5$; EXAONE-Deep-32B: $+0.56$, $p_{\mathrm{LR}} = 0.038$), indicating that increasing the needle ratio attenuates length-related degradation (i.e., more signal per distractor helps especially on longer sequences). In contrast, other models show a negative and significant $N \times \rho$ (DS-Qwen-7B: $-0.81$, $p_{\mathrm{LR}} = 0.0025$; gemini-2.5-pro: $-8.61$, $p_{\mathrm{LR}} = 0.0021$), consistent with long contexts amplifying sensitivity to distractors. Several additional models exhibit borderline positive evidence ($0.05 < p_{\mathrm{LR}} < 0.10$; e.g., Qwen3-32B, Qwen3-8B).

| Model | $\beta_{d\times N}$ | $p_{\mathrm{LR}}(d\times N)$ | $\beta_{d\times\rho}$ | $p_{\mathrm{LR}}(d\times\rho)$ | $\beta_{N\times\rho}$ | $p_{\mathrm{LR}}(N\times\rho)$ | $\beta_{d\times N\times\rho}$ | $p_{\mathrm{LR}}(d\times N\times\rho)$ |
|---|---|---|---|---|---|---|---|---|
| DeepSeek-R1-0528 | $-0.03$ | $0.864$ | $0.07$ | $0.89$ | $1.41$ | $0.421$ | $-0.00$ | $0.997$ |
| DeepSeek-R1-Distill-Llama-70B | $-0.15$ | $3.41e-05^{***}$ | $-0.21$ | $0.0552$ | $0.28$ | $0.433$ | $0.14$ | $0.014^{*}$ |
| DeepSeek-R1-Distill-Qwen-1.5B | $-0.01$ | $0.706$ | $-0.15$ | $0.195$ | $-0.25$ | $0.402$ | $0.08$ | $0.211$ |
| DeepSeek-R1-Distill-Qwen-32B | $-0.37$ | $4.07e-32^{***}$ | $-0.79$ | $1.84e-17^{***}$ | $-0.11$ | $0.702$ | $0.36$ | $8.14e-13^{***}$ |
| DeepSeek-R1-Distill-Qwen-7B | $-0.16$ | $7.63e-07^{***}$ | $-0.51$ | $1.41e-07^{***}$ | $-0.81$ | $0.00246^{**}$ | $0.26$ | $9.41e-07^{***}$ |
| EXAONE-Deep-32B | $-0.27$ | $1.28e-17^{***}$ | $-0.89$ | $2.73e-20^{***}$ | $0.56$ | $0.0378^{*}$ | $0.42$ | $2.12e-15^{***}$ |
| Phi-4-mini-reasoning | $-0.13$ | $4.96e-05^{***}$ | $-0.62$ | $2.44e-10^{***}$ | $-0.04$ | $0.886$ | $0.30$ | $1.08e-08^{***}$ |
| Phi-4-reasoning | $-0.45$ | $1.94e-30^{***}$ | $-0.50$ | $2.66e-05^{***}$ | $0.59$ | $0.0917$ | $0.18$ | $0.00696^{**}$ |
| Phi-4-reasoning-plus | $-0.41$ | $4.28e-25^{***}$ | $-0.39$ | $0.000821^{***}$ | $0.54$ | $0.124$ | $0.15$ | $0.0199^{*}$ |
| QwQ-32B | $-0.39$ | $1.25e-33^{***}$ | $-0.96$ | $3.77e-23^{***}$ | $-0.13$ | $0.652$ | $0.36$ | $3.72e-12^{***}$ |
| Qwen3-1.7B | $-0.20$ | $7.33e-11^{***}$ | $-0.56$ | $7.48e-09^{***}$ | $0.04$ | $0.887$ | $0.27$ | $3.55e-07^{***}$ |
| Qwen3-30B-A3B | $-0.19$ | $4.57e-09^{***}$ | $-0.25$ | $0.00919^{**}$ | $1.16$ | $7.18e-05^{***}$ | $0.08$ | $0.134$ |
| Qwen3-32B | $-0.22$ | $1.76e-10^{***}$ | $-0.56$ | $6.88e-08^{***}$ | $0.53$ | $0.0965$ | $0.21$ | $0.000104^{***}$ |
| Qwen3-8B | $-0.24$ | $6.63e-14^{***}$ | $-0.46$ | $8.97e-07^{***}$ | $0.50$ | $0.0767$ | $0.21$ | $5.36e-05^{***}$ |
| gemini-2.5-flash | $-0.58$ | $0.0041^{**}$ | $-0.31$ | $0.681$ | $-3.05$ | $0.133$ | $0.16$ | $0.67$ |
| gemini-2.5-flash-lite | $-0.27$ | $0.00554^{**}$ | $-0.79$ | $0.00686^{**}$ | $-1.20$ | $0.171$ | $0.31$ | $0.0582$ |
| gemini-2.5-pro | $-1.84$ | $7.81e-08^{***}$ | $-3.53$ | $0.00601^{**}$ | $-8.61$ | $0.00211^{**}$ | $1.61$ | $0.00891^{**}$ |
| gpt-5-2025-08-07 | $-0.13$ | $0.685$ | $1.41$ | $0.276$ | $5.23$ | $0.197$ | $-1.05$ | $0.0762$ |
| gpt-5-mini-2025-08-07 | $-0.56$ | $0.00034^{***}$ | $0.01$ | $0.985$ | $0.48$ | $0.756$ | $-0.06$ | $0.838$ |
| gpt-5-nano-2025-08-07 | $-0.09$ | $0.393$ | $-0.03$ | $0.929$ | $1.82$ | $0.0603$ | $-0.10$ | $0.569$ |
| o3-2025-04-16 | $-0.19$ | $0.353$ | $1.00$ | $0.212$ | $1.72$ | $0.431$ | $-0.62$ | $0.0922$ |
| o4-mini-2025-04-16 | $-0.45$ | $0.00829^{**}$ | $0.12$ | $0.856$ | $-0.05$ | $0.979$ | $-0.07$ | $0.81$ |

Table 3: Interaction analysis across intrinsic difficulty $d$, effective length $\log_{10} N$, and needle ratio $\rho$. Coefficients ($\beta$) are from a per-model logistic GLM with all pairwise and 3-way interactions plus a quadratic main effect for $\rho$; $p_{\mathrm{LR}}$ are likelihood-ratio p-values from dropping each interaction from the full model. Stars: *** $p < 0.001$, ** $p < 0.01$, * $p < 0.05$.

## B.2 Three-way effects

A positive three-way interaction $d \times N \times \rho$ is significant in 12/22 models (e.g., DS-Qwen-32B: +0.36, $p_{\text{LR}} = 8.1\text{e}{-}13$; Phi-4-mini-reasoning: +0.30, $p_{\text{LR}} = 1.1\text{e}{-}8$; QwQ-32B: +0.36, $p_{\text{LR}} = 3.7\text{e}{-}12$). Given the quadratic main effect in $\rho$, this should be read as a moderation of the *slope* with respect to $\rho$: when tasks are simultaneously long and intrinsically complex, changes in $\rho$ tend to have a larger beneficial marginal effect (i.e., increasing the share of needle statements mitigates the compounded $d \times N$ burden more than when only one of $d$ or $N$ is high). Frontier models show near-zero or slightly negative three-way coefficients (gpt-5: $-1.05$, $p_{\text{LR}} = 0.076$; o3: $-0.62$, $p_{\text{LR}} = 0.092$), aligning with their weak $\rho$-dependence overall.

## B.3 Practical implications

Beyond main-effect sensitivities, many models exhibit genuine interaction structure: high $d$ and high $N$ *jointly* depress accuracy more than additively, and high $d$ *amplifies* distractor harm ($d \times \rho < 0$). For a substantial subset of models, increasing the needle share ($\rho$) yields greater gains specifically when tasks are both long and complex (positive $d \times N \times \rho$), providing a concrete knob for mitigating compounded failure modes. It is important to note however that for models with long-context budget issues (e.g., gemini-2.5-pro; see Appendix Appendix H), $N \times \rho$ estimates can be dominated by those failures, inflating effect sizes.

## C  Model performance with Error-Bars

Due the small size and the large amount of models we compare in Figure 1 of the main paper we omit error-bars to improve legibility. In this section we plot larger versions of the charts in the panel with vertical error bars representing 90% confidence intervals for the mean accuracy of each model–condition pair.

We compute these confidence intervals with the Wilson score method for a binomial proportion. Specifically, for a given point with $k$ correct answers out of $n$ trials we set $\hat{p} = k/n$ and use the 90 % standard-normal quantile $z = 1.644853627$ to obtain

$$[l,\ u] \;=\; \frac{\hat{p} + \frac{z^2}{2n} \;\pm\; z\sqrt{\dfrac{\hat{p}\,(1-\hat{p}) + z^2/(4n)}{n}}}{1 + \frac{z^2}{n}}.$$

The resulting interval $[l, u]$ marks the range that would contain the true underlying accuracy in 90% of repeated experiments with the same sample size. Wider bars correspond to greater sampling uncertainty, whereas tighter bars indicate more stable estimates.

The narrow error bars we observe substantiate the discussion of the results in the main paper and highlights a significant difference between the curves at 90% significance and the presence of the characteristic U-shape.

## D  Evaluation, scoring pipeline, and error taxonomy

This section fully details the evaluation and scoring pipeline functionally in text and in pseudo code.

### D.1  Inputs, files, and naming

The scorer consumes three artefacts per puzzle instance $z$:

1. Model output of the reasoning trace + response (plain text).

2. Puzzle text (for the question): The last line is the natural-language question.

3. Puzzle metadata (JSON state): providing the person-of-interest (PoI), gold label, needles, difficulty, statement count, and category domains.

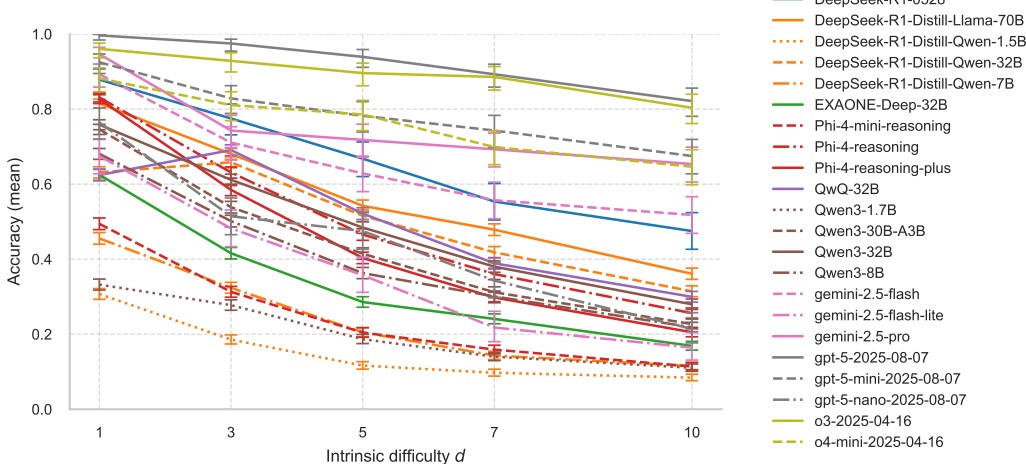

Figure 5: Accuracy relative to difficulty $d$ for $d \in \{1, 3, 5, 7, 10\}$ on the X-Axis

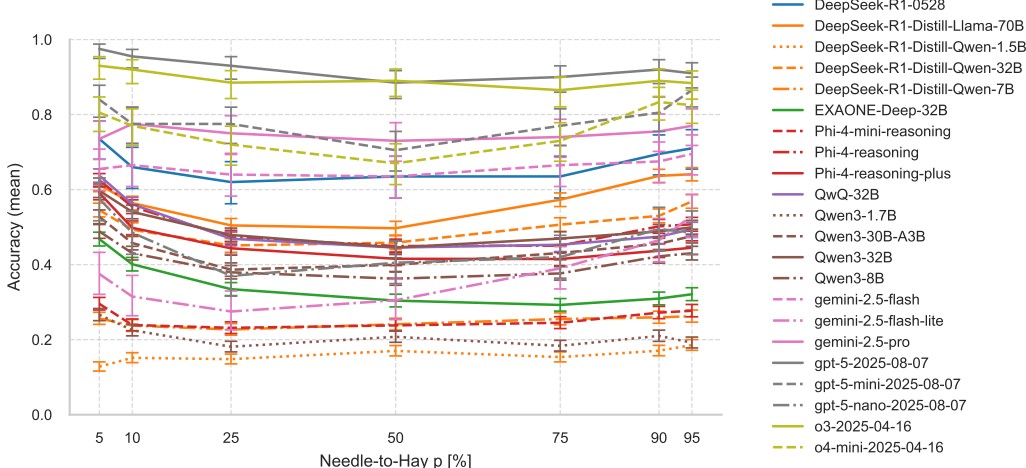

Figure 6: Accuracy relative to distractor-ratio $\rho$ for $\rho \in \{5, 10, 25, 50, 75, 90, 95\}$ on the X-Axis

## D.2 CATEGORY MAPPING AND QUALIFIERS

The target category is resolved from the last line question using deterministic templates:

- `startswith("Where is")` $\rightarrow$ `location`
- `startswith("What color shirt")` $\rightarrow$ `clothes_shirt`
- `startswith("What color pant")` $\rightarrow$ `clothes_pant`
- `startswith("What color hat")` $\rightarrow$ `clothes_hat`
- `startswith("What color of socks")` $\rightarrow$ `clothes_socks`
- `startswith("What color of gloves")` $\rightarrow$ `clothes_gloves`
- `startswith("What color of underwear")` $\rightarrow$ `clothes_underwear`
- `startswith("What is the final hair color")` $\rightarrow$ `hair`
- `endswith("most recently eat?")` $\rightarrow$ `recent_eat`
- `endswith("recently watch?")` $\rightarrow$ `recent_watch`

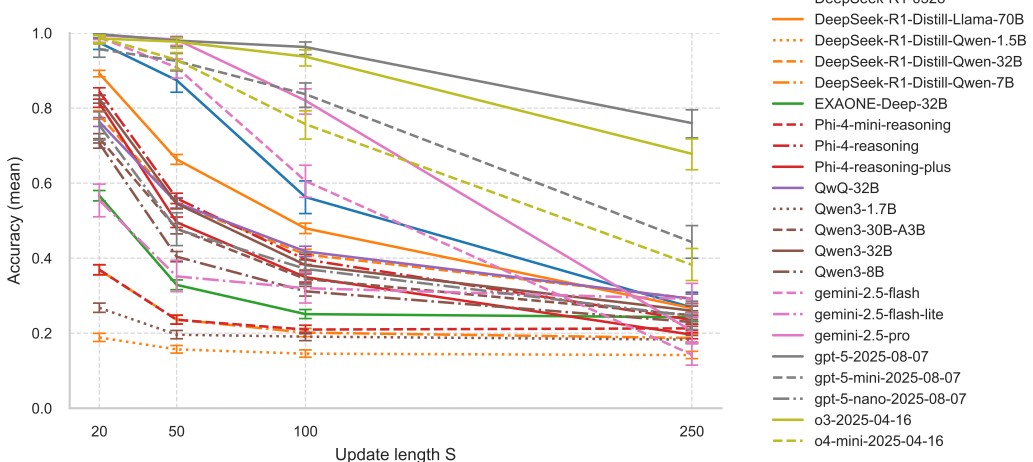

Figure 7: Accuracy relative to total statement length $N$ for $N \in \{20, 50, 100, 250\}$ on the X-Axis

- endswith("recently listen to?") → recent_listen
- endswith("recently read?") → recent_read

Qualifiers (used to find a "valid PoI sentence"):

- location: {at, located, in}
- clothes_shirt: {shirt, wear}; clothes_pant: {pant, wear}; clothes_hat: {hat, wear}; clothes_gloves: {glove, wear}; clothes_socks: {sock, wear}; clothes_underwear: {underwear, wear}
- hair: {hair}
- recent_eat: {eat, ate}; recent_watch: {watch, watched, movie}; recent_listen: {listen, listened, music}; recent_read: {read, book}

The PoI name and the domain values for the target category are read from the puzzle metadata file.

### D.3 NORMALIZATION, SENTENCE WINDOWS, AND CONTEXT BUDGET

Normalization:

- Lowercase the full model output; split by newlines.
- If the last line starts with "(" and ends with ")", drop that line because it's usually just a clarification that follows the final answer.
- Drop empty lines; if no lines remain, keep a single empty line.

Sentence windows:

1. **valid_poi_last_sentence**: last line containing PoI and any category qualifier; take `TakeLastSentence`.
2. **poi_last_sentence**: last line containing PoI; take `TakeLastSentence` of that line.
3. **last_sentence**: apply `TakeLastSentence` to the last remaining line; see below.

TakeLastSentence($s$): let $p = \text{split}(s, ".")$. If $\text{len}(p) \geq 2$, return $p[-2]$; else return $s$.

Context limit failure: we mark `wrong_max_context` if any of the following holds:

$$\text{prompt\_tokens} + \text{response\_tokens} + 20 \geq 32'768, \quad \text{or} \quad \text{last\_sentence} = "", \quad \text{or} \quad \text{empty output}.$$

The 20 additional tokens added to the `response_tokens` is necessary because when models usually produce incomplete responses a few tokens before the maximum-output-token limit they were prompted to decode with.

## D.4 MATCHING, SYNONYMS, AND ALTERNATIVES

Since some models respond with the answer attribute starting with a special character instead of an empty space (like a word) the containment accepts the following alternative word boundries (code `CheckAnswer(term, sentence)` returns true if any holds):

- `sentence.startswith(term)` or `sentence == term`
- substring appears prefixed by one of: space, "[", """, "*", "_", "{ ", "("

Since some attributes in the attribute taxonomy could be spelled alternatively, we relax Synonym whitelist for them in the evaluation (code `CheckCorrectAnswer(gold, sentence)`):

- `reaggea`: accept `reaggea` or `reggae`
- `sci-fi`: accept `sci-fi`, `science fiction`, `science-fiction`
- `camp`: accept `camp`, `campground`
- `potatoes`: accept `potatoes`, `potato`
- `market`: accept `market`, `marketplace`
- `livingroom`: accept `livingroom`, `living room`
- otherwise: accept exactly `gold` by boundary-tolerant containment

We further ensure no alternative value is present in the last sentence and therefore compute (`AltFlag(sentence, alts, gold)`) as:

1. Let $s$ = lowercase(sentence), $g$ = lowercase(gold), and $A$ = lowercase alternatives = domain values $\setminus \{g\}$.
2. Define $present(t) = $ `CheckCorrectAnswer`$(t, s)$.
3. If no alternative is present, return false.
4. If both gold and at least one alternative are present, keep the alternative flag only if some alternative's character span encloses the gold's span in $s$ ("spans" test); otherwise clear the flag. This is done to accommodate attribute values like 'non-fiction' which contain alternative values like 'fiction'.
5. Since sometimes models respond by mentioning also previous value of the attribute (eg a sentence like `"... the final step explicitly converts gray to green, resulting in **green**"`) we do not set the flag if the gold flag is the last mentioned attribute in the sentence.

## D.5 BUCKETS, PRECEDENCE, AND ACCURACY

Let:

$$
\begin{aligned}
c_{\text{valid}} &= \texttt{CheckCorrectAnswer}(\text{gold}, \texttt{valid\_poi\_last\_sentence}), \\
c_{\text{poi}} &= \texttt{CheckCorrectAnswer}(\text{gold}, \texttt{poi\_last\_sentence}), \\
c_{\text{last}} &= \texttt{CheckCorrectAnswer}(\text{gold}, \texttt{last\_sentence}), \\
a_{\text{valid}} &= \texttt{AltFlag}(\texttt{valid\_poi\_last\_sentence}, \texttt{alts}, \texttt{gold}), \\
a_{\text{poi}} &= \texttt{AltFlag}(\texttt{poi\_last\_sentence}, \texttt{alts}, \texttt{gold}), \\
a_{\text{last}} &= \texttt{AltFlag}(\texttt{last\_sentence}, \texttt{alts}, \texttt{gold}).
\end{aligned}
$$

Classification (first applicable wins):

1. if $c_{\text{valid}}$ and not $a_{\text{valid}}$: `correct_valid`

2. else if $c_{\text{poi}}$ and not $a_{\text{poi}}$: `correct_poi`

3. else if $c_{\text{last}}$ and not $a_{\text{last}}$: `correct_last_sentence`

4. else if not $c_{\text{valid}}$ and $a_{\text{valid}}$ and `valid_poi_last_sentence` $\neq$ `""`: `wrong_logic`

5. else if not $c_{\text{poi}}$ and $a_{\text{poi}}$ and `poi_last_sentence` $\neq$ `""`: `wrong_logic_poi`

6. else if not $c_{\text{last}}$ and $a_{\text{last}}$ and `last_sentence` $\neq$ `""`: `wrong_logic_last_sentence`

7. else: `wrong_other`

Note: When a sentence contains both gold and an alternative that spans the gold, this does not trigger a `wrong_logic*` bucket unless the corresponding "not $c\_\cdot$ and $a\_\cdot$" guard holds; such cases can fall through to `wrong_other`, matching the implementation. The main figures/regressions of the paper use accuracy with correct $\in$ {`correct_valid, correct_poi, correct_last_sentence`} and correct as the complement.

## D.6 PSEUDOCODE

Using pseudo code, figure 1 explains the data preparation, figure 2 explains the scoring of the results and success/failure types, and figure 3 details the used utility functions.

---

**Algorithm 1:** Pseudocode of the data preparation.

```
// Parse identifiers and gold from puzzle name:
   N_d_needles_puzzleId_gold
```
$(N, d, \text{needles}, \text{puzzleId}, \text{gold}) \leftarrow \text{PARSE}(\texttt{ans\_path})$;
```
// Load question and resolve category
```
$\text{question} \leftarrow \text{LAST\_LINE}(\texttt{puzzle\_txt\_path})$;
$\text{category} \leftarrow \text{MapQuestionToCategory}(\text{question})$;
```
// Load metadata needed for evaluation
```
$(\text{poi}, \text{domains}) \leftarrow \text{READJSON}(\texttt{state\_json\_path})$;
$\text{poi} \leftarrow \text{LOWER}(\text{poi})$; $\text{domainValues} \leftarrow \text{domains}[\text{category}]$;
$\text{quals} \leftarrow \text{GetQualifiersFor}(\text{category})$;
```
// Read and normalize model output
```
$\text{resp} \leftarrow \text{READ}(\texttt{ans\_path})$;
$\text{lines} \leftarrow \text{SPLIT\_LINES}(\text{LOWER}(\text{resp}))$;
**if** $\text{STARTSWITH}(lines[-1], "(") \wedge \text{ENDSWITH}(lines[-1], ")")$ **then**
  $\lfloor$ $\text{DROP\_LAST}(\text{lines})$

$\text{lines} \leftarrow \text{FILTER\_NONEMPTY}(\text{lines})$;
**if** $\text{LEN}(lines) = 0$ **then**
  $\lfloor$ $\text{lines} \leftarrow [""]$

---

## E ERROR DISTRIBUTION

The following tables illustrate the error-type distribution per difficulty 4, and length 5. The purpose of this distinction is to illustrate three primary error types (i.e., long context error, logic error, other error) and to distinguish among the logic errors how well the model followed the instruction in its response formatting. The other error captures cases where the model responds but doesn't mention any attribute, usually this is cases where the model erroneously assumes the problem can't be solved with the provided information. For conciseness the error types in the table headers are abbreviated as following:

- Wrong max-context → ctx
- Wrong logic (last valid POI sentence) → valid-logic

**Algorithm 2:** Pseudocode of the CogniLoad scoring pipeline.

```
// Prepare data
// Extract windows
last_sentence ← TakeLastSentence(lines[−1]);
poi_lines ← {s ∈ lines : CONTAINS(s, poi)};
poi_last ← LASTOREMPTY(poi_lines);
valid_poi_lines ← {s ∈ poi_lines : CONTAINSANY(s, quals)};
valid_poi_last ← LASTOREMPTY(valid_poi_lines);
// Budget/emptiness guard
```
**if** ExceededContext($runtime\_meta$) $\lor$ $last\_sentence$ = ”” $\lor$ LEN($resp$) = 0 **then**
  **return** wrong_max_context
```
// Gold matches
```
$c_{\text{valid}}$ ← CheckCorrectAnswer(gold, valid_poi_last);
$c_{\text{poi}}$ ← CheckCorrectAnswer(gold, poi_last);
$c_{\text{last}}$ ← CheckCorrectAnswer(gold, last_sentence);
```
// Alternative flags with span rule
```
alts ← {$v \in$ domainValues : $v \neq$ gold};
$a_{\text{valid}}$ ← AltFlag(valid_poi_last, alts, gold);
$a_{\text{poi}}$ ← AltFlag(poi_last, alts, gold);
$a_{\text{last}}$ ← AltFlag(last_sentence, alts, gold);
```
// Buckets (first match wins)
```
**if** $c_{\text{valid}} \land \neg a_{\text{valid}}$ **then**
  **return** correct_valid
**else if** $c_{\text{poi}} \land \neg a_{\text{poi}}$ **then**
  **return** correct_poi
**else if** $c_{\text{last}} \land \neg a_{\text{last}}$ **then**
  **return** correct_last_sentence
**else if** $\neg c_{\text{valid}} \land a_{\text{valid}} \land valid\_poi\_last \neq$ ”” **then**
  **return** wrong_logic
**else if** $\neg c_{\text{poi}} \land a_{\text{poi}} \land poi\_last \neq$ ”” **then**
  **return** wrong_logic_poi
**else if** $\neg c_{\text{last}} \land a_{\text{last}} \land last\_sentence \neq$ ”” **then**
  **return** wrong_logic_last_sentence
**else**
  **return** wrong_other

- Wrong logic (last POI sentence) → poi-logic
- Wrong logic (last sentence) → last-logic
- Wrong other → other

If the total number of evaluated samples does not precisely match the expected number of samples (e.g 'o3-2025-04-16' with difficulty 5 resulted in 279 vs 280 expected examples), this usually indicates the puzzle was not processed due to a an API error (e.g. a security refusal). These few missing cases were attempted to be evaluated a second time but refused again.

Table 4: Error-type distribution by difficulty $d$ per model (aggregated over $N$ and $\rho$).

| Model | Dim | Samples | Acc [%] | ctx | valid-logic | poi-logic | last-logic | other |
|---|---|---|---|---|---|---|---|---|
| DeepSeek-R1-0528 | 1 | 280 | 88 | 3 | 31 | 0 | 0 | 0 |
| DeepSeek-R1-0528 | 3 | 280 | 78 | 11 | 52 | 0 | 0 | 0 |
| DeepSeek-R1-0528 | 5 | 280 | 67 | 13 | 80 | 0 | 0 | 0 |
| DeepSeek-R1-0528 | 7 | 280 | 55 | 16 | 109 | 0 | 0 | 0 |
| DeepSeek-R1-0528 | 10 | 280 | 48 | 19 | 128 | 0 | 0 | 0 |

Table 4: Error-type distribution by difficulty $d$ per model (aggregated over $N$ and $rho$). (continued)

| Model | Dim | Samples | Acc [%] | ctx | valid-logic | poi-logic | last-logic | other |
|---|---|---|---|---|---|---|---|---|
| DeepSeek-R1-Distill-Llama-70B | 1 | 2800 | 82 | 0 | 507 | 0 | 2 | 9 |
| DeepSeek-R1-Distill-Llama-70B | 3 | 2800 | 68 | 11 | 877 | 0 | 3 | 3 |
| DeepSeek-R1-Distill-Llama-70B | 5 | 2800 | 54 | 12 | 1260 | 0 | 8 | 2 |
| DeepSeek-R1-Distill-Llama-70B | 7 | 2800 | 48 | 28 | 1424 | 0 | 4 | 4 |
| DeepSeek-R1-Distill-Llama-70B | 10 | 2800 | 36 | 33 | 1747 | 0 | 4 | 5 |
| DeepSeek-R1-Distill-Qwen-1.5B | 1 | 2800 | 31 | 561 | 1140 | 4 | 81 | 154 |
| DeepSeek-R1-Distill-Qwen-1.5B | 3 | 2800 | 19 | 543 | 1305 | 3 | 82 | 347 |
| DeepSeek-R1-Distill-Qwen-1.5B | 5 | 2800 | 12 | 613 | 1381 | 4 | 63 | 413 |
| DeepSeek-R1-Distill-Qwen-1.5B | 7 | 2800 | 10 | 695 | 1323 | 7 | 60 | 443 |
| DeepSeek-R1-Distill-Qwen-1.5B | 10 | 2800 | 8 | 717 | 1371 | 7 | 62 | 407 |
| DeepSeek-R1-Distill-Qwen-32B | 1 | 2800 | 63 | 1 | 1023 | 0 | 3 | 5 |
| DeepSeek-R1-Distill-Qwen-32B | 3 | 2800 | 66 | 46 | 908 | 0 | 2 | 1 |
| DeepSeek-R1-Distill-Qwen-32B | 5 | 2800 | 52 | 78 | 1275 | 0 | 1 | 2 |
| DeepSeek-R1-Distill-Qwen-32B | 7 | 2800 | 42 | 94 | 1532 | 0 | 0 | 3 |
| DeepSeek-R1-Distill-Qwen-32B | 10 | 2800 | 31 | 106 | 1813 | 0 | 0 | 1 |
| DeepSeek-R1-Distill-Qwen-7B | 1 | 2800 | 46 | 47 | 1454 | 5 | 4 | 15 |
| DeepSeek-R1-Distill-Qwen-7B | 3 | 2800 | 32 | 124 | 1561 | 5 | 70 | 134 |
| DeepSeek-R1-Distill-Qwen-7B | 5 | 2800 | 20 | 246 | 1706 | 3 | 65 | 206 |
| DeepSeek-R1-Distill-Qwen-7B | 7 | 2800 | 14 | 317 | 1802 | 8 | 49 | 223 |
| DeepSeek-R1-Distill-Qwen-7B | 10 | 2800 | 12 | 352 | 1770 | 6 | 60 | 290 |
| EXAONE-Deep-32B | 1 | 2800 | 62 | 0 | 995 | 5 | 2 | 48 |
| EXAONE-Deep-32B | 3 | 2800 | 42 | 2 | 1595 | 2 | 27 | 10 |
| EXAONE-Deep-32B | 5 | 2800 | 29 | 10 | 1938 | 2 | 45 | 5 |
| EXAONE-Deep-32B | 7 | 2800 | 24 | 14 | 2060 | 1 | 42 | 9 |
| EXAONE-Deep-32B | 10 | 2800 | 17 | 16 | 2257 | 1 | 40 | 13 |
| gemini-2.5-flash | 1 | 280 | 89 | 6 | 24 | 0 | 0 | 0 |
| gemini-2.5-flash | 3 | 280 | 71 | 64 | 17 | 0 | 0 | 0 |
| gemini-2.5-flash | 5 | 280 | 63 | 93 | 11 | 0 | 0 | 0 |
| gemini-2.5-flash | 7 | 280 | 56 | 116 | 8 | 0 | 0 | 0 |
| gemini-2.5-flash | 10 | 280 | 52 | 110 | 25 | 0 | 0 | 0 |
| gemini-2.5-flash-lite | 1 | 280 | 68 | 0 | 91 | 0 | 0 | 0 |
| gemini-2.5-flash-lite | 3 | 280 | 48 | 5 | 138 | 0 | 0 | 2 |
| gemini-2.5-flash-lite | 5 | 280 | 36 | 3 | 176 | 0 | 0 | 1 |
| gemini-2.5-flash-lite | 7 | 280 | 22 | 8 | 209 | 0 | 0 | 2 |
| gemini-2.5-flash-lite | 10 | 280 | 16 | 1 | 227 | 2 | 0 | 4 |
| gemini-2.5-pro | 1 | 280 | 95 | 10 | 5 | 0 | 0 | 0 |
| gemini-2.5-pro | 3 | 280 | 74 | 69 | 3 | 0 | 0 | 0 |
| gemini-2.5-pro | 5 | 280 | 72 | 71 | 8 | 0 | 0 | 0 |
| gemini-2.5-pro | 7 | 280 | 69 | 79 | 7 | 0 | 0 | 0 |
| gemini-2.5-pro | 10 | 280 | 65 | 88 | 9 | 0 | 0 | 0 |
| gpt-5-2025-08-07 | 1 | 280 | 100 | 0 | 1 | 0 | 0 | 0 |
| gpt-5-2025-08-07 | 3 | 280 | 98 | 0 | 7 | 0 | 0 | 0 |
| gpt-5-2025-08-07 | 5 | 280 | 94 | 0 | 17 | 0 | 0 | 0 |
| gpt-5-2025-08-07 | 7 | 280 | 89 | 3 | 26 | 0 | 0 | 1 |
| gpt-5-2025-08-07 | 10 | 280 | 82 | 29 | 21 | 0 | 0 | 0 |
| gpt-5-mini-2025-08-07 | 1 | 280 | 92 | 0 | 14 | 0 | 0 | 7 |
| gpt-5-mini-2025-08-07 | 3 | 280 | 83 | 0 | 46 | 0 | 0 | 2 |
| gpt-5-mini-2025-08-07 | 5 | 280 | 78 | 3 | 52 | 0 | 1 | 5 |
| gpt-5-mini-2025-08-07 | 7 | 280 | 74 | 12 | 58 | 0 | 0 | 2 |
| gpt-5-mini-2025-08-07 | 10 | 280 | 68 | 27 | 63 | 0 | 0 | 1 |
| gpt-5-nano-2025-08-07 | 1 | 280 | 76 | 0 | 65 | 1 | 0 | 0 |
| gpt-5-nano-2025-08-07 | 3 | 280 | 51 | 0 | 133 | 0 | 3 | 0 |
| gpt-5-nano-2025-08-07 | 5 | 280 | 48 | 0 | 145 | 0 | 2 | 0 |
| gpt-5-nano-2025-08-07 | 7 | 280 | 34 | 0 | 183 | 0 | 1 | 0 |

Table 4: Error-type distribution by difficulty $d$ per model (aggregated over $N$ and $rho$). (continued)

| Model | Dim | Samples | Acc [%] | ctx | valid-logic | poi-logic | last-logic | other |
|---|---|---|---|---|---|---|---|---|
| gpt-5-nano-2025-08-07 | 10 | 280 | 21 | 0 | 219 | 0 | 1 | 0 |
| o3-2025-04-16 | 1 | 279 | 96 | 0 | 11 | 0 | 0 | 0 |
| o3-2025-04-16 | 3 | 280 | 93 | 0 | 20 | 0 | 0 | 0 |
| o3-2025-04-16 | 5 | 279 | 90 | 1 | 28 | 0 | 0 | 0 |
| o3-2025-04-16 | 7 | 280 | 89 | 2 | 30 | 0 | 0 | 0 |
| o3-2025-04-16 | 10 | 280 | 80 | 21 | 34 | 0 | 0 | 0 |
| o4-mini-2025-04-16 | 1 | 279 | 88 | 0 | 31 | 0 | 0 | 2 |
| o4-mini-2025-04-16 | 3 | 280 | 81 | 0 | 53 | 0 | 0 | 0 |
| o4-mini-2025-04-16 | 5 | 280 | 79 | 3 | 57 | 0 | 0 | 0 |
| o4-mini-2025-04-16 | 7 | 279 | 70 | 5 | 78 | 0 | 0 | 1 |
| o4-mini-2025-04-16 | 10 | 280 | 65 | 13 | 86 | 0 | 0 | 0 |
| Phi-4-mini-reasoning | 1 | 2800 | 49 | 4 | 1222 | 16 | 48 | 126 |
| Phi-4-mini-reasoning | 3 | 2800 | 31 | 1 | 1613 | 6 | 126 | 178 |
| Phi-4-mini-reasoning | 5 | 2800 | 20 | 12 | 1733 | 8 | 286 | 189 |
| Phi-4-mini-reasoning | 7 | 2800 | 16 | 7 | 1785 | 6 | 354 | 203 |
| Phi-4-mini-reasoning | 10 | 2800 | 11 | 18 | 1856 | 5 | 415 | 187 |
| Phi-4-reasoning | 1 | 2800 | 83 | 14 | 417 | 0 | 0 | 42 |
| Phi-4-reasoning | 3 | 2800 | 63 | 79 | 921 | 0 | 0 | 33 |
| Phi-4-reasoning | 5 | 2800 | 47 | 246 | 1219 | 1 | 0 | 30 |
| Phi-4-reasoning | 7 | 2800 | 36 | 522 | 1240 | 0 | 0 | 29 |
| Phi-4-reasoning | 10 | 2800 | 26 | 857 | 1200 | 1 | 0 | 25 |
| Phi-4-reasoning-plus | 1 | 2800 | 83 | 100 | 347 | 1 | 0 | 37 |
| Phi-4-reasoning-plus | 3 | 2800 | 58 | 182 | 938 | 0 | 0 | 43 |
| Phi-4-reasoning-plus | 5 | 2800 | 40 | 471 | 1153 | 1 | 0 | 46 |
| Phi-4-reasoning-plus | 7 | 2800 | 30 | 936 | 1001 | 2 | 0 | 27 |
| Phi-4-reasoning-plus | 10 | 2800 | 21 | 1198 | 994 | 0 | 1 | 32 |
| Qwen3-1.7B | 1 | 2800 | 33 | 1 | 1840 | 1 | 11 | 17 |
| Qwen3-1.7B | 3 | 2800 | 28 | 0 | 2010 | 0 | 2 | 11 |
| Qwen3-1.7B | 5 | 2800 | 19 | 0 | 2268 | 0 | 1 | 8 |
| Qwen3-1.7B | 7 | 2800 | 14 | 0 | 2396 | 0 | 2 | 10 |
| Qwen3-1.7B | 10 | 2798 | 11 | 0 | 2483 | 0 | 1 | 5 |
| Qwen3-30B-A3B | 1 | 2800 | 75 | 18 | 652 | 3 | 0 | 38 |
| Qwen3-30B-A3B | 3 | 2800 | 54 | 48 | 1237 | 1 | 0 | 6 |
| Qwen3-30B-A3B | 5 | 2800 | 41 | 40 | 1587 | 2 | 0 | 10 |
| Qwen3-30B-A3B | 7 | 2800 | 31 | 25 | 1894 | 2 | 0 | 6 |
| Qwen3-30B-A3B | 10 | 2800 | 23 | 19 | 2133 | 3 | 1 | 8 |
| Qwen3-32B | 1 | 2800 | 76 | 0 | 626 | 0 | 1 | 49 |
| Qwen3-32B | 3 | 2800 | 61 | 1 | 1075 | 1 | 1 | 10 |
| Qwen3-32B | 5 | 2800 | 48 | 0 | 1434 | 1 | 3 | 6 |
| Qwen3-32B | 7 | 2800 | 38 | 0 | 1721 | 0 | 6 | 9 |
| Qwen3-32B | 10 | 2800 | 28 | 1 | 1994 | 0 | 12 | 10 |
| Qwen3-8B | 1 | 2800 | 68 | 77 | 739 | 2 | 0 | 76 |
| Qwen3-8B | 3 | 2800 | 50 | 136 | 1190 | 3 | 18 | 49 |
| Qwen3-8B | 5 | 2800 | 36 | 106 | 1619 | 3 | 19 | 37 |
| Qwen3-8B | 7 | 2800 | 30 | 101 | 1791 | 0 | 27 | 39 |
| Qwen3-8B | 10 | 2800 | 22 | 78 | 2046 | 0 | 39 | 25 |
| QwQ-32B | 1 | 2800 | 62 | 2 | 1044 | 1 | 0 | 5 |
| QwQ-32B | 3 | 2800 | 69 | 12 | 850 | 0 | 3 | 1 |
| QwQ-32B | 5 | 2800 | 52 | 52 | 1279 | 0 | 3 | 6 |
| QwQ-32B | 7 | 2800 | 39 | 112 | 1596 | 0 | 1 | 2 |
| QwQ-32B | 10 | 2800 | 30 | 273 | 1674 | 0 | 5 | 9 |

---

**Algorithm 3:** Pseudocode of the CogniLoad scoring pipeline utility functions.

---

**Utilities.**

**ExceededContext**: return true if `prompt_tokens + response_tokens + 20 ≥ 32 768`.

**TakeLastSentence**(s): let $p = $ SPLIT$(s,"."). If LEN$(p) \geq 2$, return $p[-2]$; else $s$.

**CheckCorrectAnswer**(gold, s):

- **CheckAnswer**(t, s) is true if any holds: STARTSWITH$(s, t)$, $s = t$, or $s$ contains one of: "space+t", "[ t", "" t", "*t", "_t", "{ t", "(t".

- If $gold = $ reaggea: return **CheckAnswer**(reaggea) $\vee$ **CheckAnswer**(reggae).

- If sci-fi: return
  **CheckAnswer**(sci-fi) $\vee$ **CheckAnswer**(science fiction) $\vee$ **CheckAnswer**(science-fiction).

- If camp: **CheckAnswer**(camp) $\vee$ **CheckAnswer**(campground).

- If potatoes: **CheckAnswer**(potatoes) $\vee$ **CheckAnswer**(potato).

- If market: **CheckAnswer**(market) $\vee$ **CheckAnswer**(marketplace).

- If livingroom: **CheckAnswer**(livingroom) $\vee$ **CheckAnswer**(living room).

- Else: **CheckAnswer**(gold).

**AltFlag**(s, alts, gold):

1. $s \leftarrow$ LOWER$(s)$, $g \leftarrow$ LOWER$(gold)$, $A \leftarrow \{$LOWER$(a) : a \in$ alts$, a \neq g\}$.

2. alt_present $\leftarrow \bigvee_{a \in A}$ **CheckCorrectAnswer**$(a, s)$.

3. $g\_idx \leftarrow s$.RFIND$(g)$.

4. **if** $g\_idx \neq -1$ and alt_present: let $g\_end = g\_idx +$ LEN$(g)$; set spans =false. For each $a \in A$: $a\_idx \leftarrow s$.RFIND$(a)$; if $a\_idx \neq -1$ and $a\_idx \leq g\_idx$ and $a\_idx +$ LEN$(a) \geq g\_end$, set spans =true. If ¬spans, set alt_present =false.

5. Return alt_present.

---

Table 5: Error-type distribution by puzzle length $N$ per model (aggregated over $d$ and $\rho$).

| Model | Dim | Samples | Acc [%] | ctx | valid-logic | poi-logic | last-logic | other |
|---|---|---|---|---|---|---|---|---|
| DeepSeek-R1-0528 | 20 | 350 | 97 | 7 | 2 | 0 | 0 | 0 |
| DeepSeek-R1-0528 | 50 | 350 | 87 | 14 | 30 | 0 | 0 | 0 |
| DeepSeek-R1-0528 | 100 | 350 | 56 | 25 | 128 | 0 | 0 | 0 |
| DeepSeek-R1-0528 | 250 | 350 | 27 | 16 | 240 | 0 | 0 | 0 |
| DeepSeek-R1-Distill-Llama-70B | 20 | 3500 | 89 | 0 | 372 | 0 | 1 | 4 |
| DeepSeek-R1-Distill-Llama-70B | 50 | 3500 | 66 | 2 | 1167 | 0 | 4 | 5 |
| DeepSeek-R1-Distill-Llama-70B | 100 | 3500 | 48 | 2 | 1811 | 0 | 6 | 3 |
| DeepSeek-R1-Distill-Llama-70B | 250 | 3500 | 27 | 80 | 2465 | 0 | 10 | 11 |
| DeepSeek-R1-Distill-Qwen-1.5B | 20 | 3500 | 19 | 749 | 1676 | 8 | 58 | 349 |
| DeepSeek-R1-Distill-Qwen-1.5B | 50 | 3500 | 16 | 752 | 1749 | 5 | 85 | 360 |
| DeepSeek-R1-Distill-Qwen-1.5B | 100 | 3500 | 15 | 698 | 1749 | 4 | 90 | 450 |
| DeepSeek-R1-Distill-Qwen-1.5B | 250 | 3500 | 14 | 930 | 1346 | 8 | 115 | 605 |
| DeepSeek-R1-Distill-Qwen-32B | 20 | 3500 | 78 | 1 | 760 | 0 | 0 | 2 |
| DeepSeek-R1-Distill-Qwen-32B | 50 | 3500 | 55 | 1 | 1583 | 0 | 2 | 3 |
| DeepSeek-R1-Distill-Qwen-32B | 100 | 3500 | 41 | 0 | 2061 | 0 | 1 | 3 |
| DeepSeek-R1-Distill-Qwen-32B | 250 | 3500 | 29 | 323 | 2147 | 0 | 3 | 4 |
| DeepSeek-R1-Distill-Qwen-7B | 20 | 3500 | 37 | 117 | 2012 | 4 | 14 | 63 |
| DeepSeek-R1-Distill-Qwen-7B | 50 | 3500 | 24 | 207 | 2298 | 1 | 43 | 123 |
| DeepSeek-R1-Distill-Qwen-7B | 100 | 3500 | 20 | 311 | 2177 | 11 | 75 | 221 |
| DeepSeek-R1-Distill-Qwen-7B | 250 | 3500 | 19 | 451 | 1806 | 11 | 116 | 461 |
| EXAONE-Deep-32B | 20 | 3500 | 57 | 7 | 1451 | 4 | 35 | 20 |
| EXAONE-Deep-32B | 50 | 3500 | 33 | 17 | 2255 | 2 | 56 | 19 |
| EXAONE-Deep-32B | 100 | 3500 | 25 | 7 | 2556 | 4 | 34 | 21 |

Table 5: Error-type distribution by puzzle length $N$ per model (aggregated over $d$ and $\rho$). (continued)

| Model | Dim | Samples | Acc [%] | ctx | valid-logic | poi-logic | last-logic | other |
|---|---|---|---|---|---|---|---|---|
| EXAONE-Deep-32B | 250 | 3500 | 24 | 11 | 2583 | 1 | 31 | 25 |
| gemini-2.5-flash | 20 | 350 | 99 | 0 | 4 | 0 | 0 | 0 |
| gemini-2.5-flash | 50 | 350 | 91 | 10 | 22 | 0 | 0 | 0 |
| gemini-2.5-flash | 100 | 350 | 61 | 99 | 39 | 0 | 0 | 0 |
| gemini-2.5-flash | 250 | 350 | 14 | 280 | 20 | 0 | 0 | 0 |
| gemini-2.5-flash-lite | 20 | 350 | 55 | 0 | 154 | 0 | 0 | 2 |
| gemini-2.5-flash-lite | 50 | 350 | 35 | 0 | 223 | 1 | 0 | 3 |
| gemini-2.5-flash-lite | 100 | 350 | 32 | 1 | 235 | 0 | 0 | 2 |
| gemini-2.5-flash-lite | 250 | 350 | 29 | 16 | 229 | 1 | 0 | 2 |
| gemini-2.5-pro | 20 | 350 | 99 | 0 | 2 | 0 | 0 | 0 |
| gemini-2.5-pro | 50 | 350 | 98 | 1 | 5 | 0 | 0 | 0 |
| gemini-2.5-pro | 100 | 350 | 82 | 48 | 15 | 0 | 0 | 0 |
| gemini-2.5-pro | 250 | 350 | 21 | 268 | 10 | 0 | 0 | 0 |
| gpt-5-2025-08-07 | 20 | 350 | 100 | 0 | 1 | 0 | 0 | 0 |
| gpt-5-2025-08-07 | 50 | 350 | 98 | 0 | 7 | 0 | 0 | 0 |
| gpt-5-2025-08-07 | 100 | 350 | 96 | 0 | 12 | 0 | 0 | 1 |
| gpt-5-2025-08-07 | 250 | 350 | 76 | 32 | 52 | 0 | 0 | 0 |
| gpt-5-mini-2025-08-07 | 20 | 350 | 96 | 0 | 12 | 0 | 0 | 3 |
| gpt-5-mini-2025-08-07 | 50 | 350 | 93 | 0 | 23 | 0 | 0 | 3 |
| gpt-5-mini-2025-08-07 | 100 | 350 | 84 | 0 | 54 | 0 | 0 | 3 |
| gpt-5-mini-2025-08-07 | 250 | 350 | 44 | 42 | 144 | 0 | 1 | 8 |
| gpt-5-nano-2025-08-07 | 20 | 350 | 75 | 0 | 86 | 0 | 0 | 0 |
| gpt-5-nano-2025-08-07 | 50 | 350 | 48 | 0 | 181 | 1 | 1 | 0 |
| gpt-5-nano-2025-08-07 | 100 | 350 | 37 | 0 | 217 | 0 | 3 | 0 |
| gpt-5-nano-2025-08-07 | 250 | 350 | 25 | 0 | 261 | 0 | 3 | 0 |
| o3-2025-04-16 | 20 | 350 | 99 | 0 | 5 | 0 | 0 | 0 |
| o3-2025-04-16 | 50 | 350 | 98 | 0 | 8 | 0 | 0 | 0 |
| o3-2025-04-16 | 100 | 350 | 94 | 0 | 22 | 0 | 0 | 0 |
| o3-2025-04-16 | 250 | 348 | 68 | 24 | 88 | 0 | 0 | 0 |
| o4-mini-2025-04-16 | 20 | 350 | 99 | 0 | 3 | 0 | 0 | 1 |
| o4-mini-2025-04-16 | 50 | 350 | 93 | 0 | 24 | 0 | 0 | 1 |
| o4-mini-2025-04-16 | 100 | 350 | 76 | 0 | 85 | 0 | 0 | 0 |
| o4-mini-2025-04-16 | 250 | 348 | 38 | 21 | 193 | 0 | 0 | 1 |
| Phi-4-mini-reasoning | 20 | 3500 | 37 | 8 | 1765 | 7 | 213 | 215 |
| Phi-4-mini-reasoning | 50 | 3500 | 24 | 4 | 2163 | 11 | 289 | 208 |
| Phi-4-mini-reasoning | 100 | 3500 | 21 | 13 | 2198 | 6 | 327 | 222 |
| Phi-4-mini-reasoning | 250 | 3500 | 21 | 17 | 2083 | 17 | 400 | 238 |
| Phi-4-reasoning | 20 | 3500 | 84 | 11 | 496 | 0 | 0 | 38 |
| Phi-4-reasoning | 50 | 3500 | 56 | 106 | 1393 | 0 | 0 | 43 |
| Phi-4-reasoning | 100 | 3500 | 40 | 332 | 1736 | 1 | 0 | 43 |
| Phi-4-reasoning | 250 | 3500 | 24 | 1269 | 1372 | 1 | 0 | 35 |
| Phi-4-reasoning-plus | 20 | 3500 | 81 | 12 | 586 | 1 | 0 | 55 |
| Phi-4-reasoning-plus | 50 | 3500 | 50 | 382 | 1326 | 3 | 0 | 54 |
| Phi-4-reasoning-plus | 100 | 3500 | 35 | 668 | 1563 | 0 | 1 | 45 |
| Phi-4-reasoning-plus | 250 | 3500 | 20 | 1825 | 958 | 0 | 0 | 31 |
| Qwen3-1.7B | 20 | 3500 | 27 | 0 | 2546 | 0 | 5 | 12 |
| Qwen3-1.7B | 50 | 3500 | 20 | 1 | 2796 | 1 | 3 | 13 |
| Qwen3-1.7B | 100 | 3500 | 19 | 0 | 2818 | 0 | 6 | 8 |
| Qwen3-1.7B | 250 | 3498 | 18 | 0 | 2837 | 0 | 3 | 18 |
| Qwen3-30B-A3B | 20 | 3500 | 72 | 25 | 942 | 1 | 1 | 12 |
| Qwen3-30B-A3B | 50 | 3500 | 48 | 40 | 1764 | 2 | 0 | 19 |
| Qwen3-30B-A3B | 100 | 3500 | 34 | 44 | 2228 | 1 | 0 | 21 |
| Qwen3-30B-A3B | 250 | 3500 | 25 | 41 | 2569 | 7 | 0 | 16 |
| Qwen3-32B | 20 | 3500 | 82 | 0 | 607 | 0 | 1 | 8 |

Table 5: Error-type distribution by puzzle length $N$ per model (aggregated over $d$ and $\rho$). (continued)

| Model | Dim | Samples | Acc [%] | ctx | valid-logic | poi-logic | last-logic | other |
|-------|-----|---------|---------|-----|-------------|-----------|------------|-------|
| Qwen3-32B | 50 | 3500 | 55 | 1 | 1577 | 0 | 3 | 10 |
| Qwen3-32B | 100 | 3500 | 38 | 0 | 2125 | 0 | 4 | 35 |
| Qwen3-32B | 250 | 3500 | 26 | 1 | 2541 | 2 | 15 | 31 |
| Qwen3-8B | 20 | 3500 | 71 | 75 | 912 | 0 | 18 | 24 |
| Qwen3-8B | 50 | 3500 | 40 | 108 | 1893 | 1 | 31 | 53 |
| Qwen3-8B | 100 | 3500 | 31 | 117 | 2198 | 3 | 27 | 64 |
| Qwen3-8B | 250 | 3500 | 23 | 198 | 2382 | 4 | 27 | 85 |
| QwQ-32B | 20 | 3500 | 76 | 8 | 810 | 0 | 4 | 7 |
| QwQ-32B | 50 | 3500 | 55 | 27 | 1550 | 0 | 1 | 6 |
| QwQ-32B | 100 | 3500 | 42 | 36 | 1991 | 0 | 5 | 5 |
| QwQ-32B | 250 | 3500 | 29 | 380 | 2092 | 1 | 2 | 5 |

## F AIC-COMPARISON

In this section we compare the Akaike Information Criterion (AIC) of the GLM model containing a pure linear $\rho$ term with a GLM model with a squared term added for $\rho$. The equations for the models are as following with $Y = 1$ indicating a correctly solved puzzle:

**Linear model**

$$\Pr(Y{=}1) = \sigma\big(\beta_0 + \beta_d\, d + \beta_N\, \log_{10} N + \beta_\rho\, \rho\big),$$

**Quadratic model**

$$\Pr(Y{=}1) = \sigma\big(\beta_0 + \beta_d\, d + \beta_N\, \log_{10} N + \beta_\rho\, \rho + \beta_{\rho^2}\, \rho^2\big),$$

To assess whether the quadratic specification provides a statistically significant improvement over the linear GLM, we compare their maximised log-likelihoods $\ell_{\text{lin}}$ and $\ell_{\text{quad}}$. The likelihood-ratio statistic $D = 2(\ell_{\text{quad}} - \ell_{\text{lin}})$ follows, under the null hypothesis that the extra term is unnecessary, a chi-squared distribution with one degree of freedom because the quadratic model introduces exactly one additional parameter.

The $p$-value reported in Table 6 is the upper-tail probability $p = \Pr(\chi_1^2 \geq D)$. p-values below $0.05$ indicate that the quadratic term yields a statistically significant gain in fit and therefore justifies its inclusion.

We find that for all except for two models (i.e., DeepSeek-R1-Distill-Qwen-1.5B and DeepSeek-R1-Distill-Qwen-7B), the quadratic term for $\rho$ results in a significantly improved AIC value. Therefore we include the quadratic $\rho$ term in the GLM specification within the paper.

## G COMPLETE ATTRIBUTE ONTOLOGY

The following section lists the full attribute ontology of all values available for the categories.

**people**  Peter, Paul, Mary, John, Mark, Jeff, Craig, Daniel, Anna, Arnoldo, Ali, Benjamin, Joe, Donald, Mitch, Chuck, Jack, Lucas, Jeniffer, Adam, Greg, Allan, David, Ellen, Fred, Hank, Hubert, Ian, Ingrid, Rebecca, Ken, Lewis, Michael, Nathaniel, Oliver, Russ, Steve, Sandy, Ted, Tanya, Veronica, Vincent, Wesley, Brad, Sam, Igor, Sue, Jan, Jeffrey, Jacques, Debby, Olivia, Benedict, Chris, Charles, Harry, Eli, Mahmoud, Chen, William, Linda, Elizabeth, Robert, Jennifer, Emily, Joseph, Thomas, Patricia, Anthony, Jessica, Brian, Lisa, Kevin, Karen, Laura, Eric, Stephanie, Michelle, George, Andrew, Joshua, Amber, Timothy, Victoria, Richard, Cynthia, Brandon, Megan, Matthew, Nancy, Jacqueline, Gary, Dorothy, Edward, Kimberly, Scott, Sara, Justin, Brittany, Ronald, Deborah, Janet, Christopher, Alexander, Samantha, Oscar, Cindy, Frank, Carl, Paula, Irene, Theresa, Dennis, Ralph, Gerald, Martin, Terry, Bryan, Lance, Corey, Casey, Brent, Derek, Travis, Austin,

| model | $p_{LR}$ | $AIC_{linear}$ | $AIC_{quad}$ |
|---|---|---|---|
| DeepSeek-R1-0528 | **0.000** | 1032.432 | **1019.177** |
| DeepSeek-R1-Distill-Llama-70B | **0.000** | 13888.760 | **13703.165** |
| DeepSeek-R1-Distill-Qwen-1.5B | 0.657 | 11582.864 | 11584.667 |
| DeepSeek-R1-Distill-Qwen-32B | **0.000** | 16487.488 | **16409.647** |
| DeepSeek-R1-Distill-Qwen-7B | **0.041** | 14227.155 | **14224.975** |
| EXAONE-Deep-32B | **0.000** | 15469.929 | **15388.288** |
| Phi-4-mini-reasoning | **0.000** | 14462.835 | **14438.087** |
| Phi-4-reasoning | **0.000** | 13344.906 | **13161.306** |
| Phi-4-reasoning-plus | **0.000** | 12603.062 | **12477.924** |
| QwQ-32B | **0.000** | 16357.613 | **16218.181** |
| Qwen3-1.7B | **0.000** | 13700.396 | **13684.334** |
| Qwen3-30B-A3B | **0.000** | 15363.614 | **15252.219** |
| Qwen3-32B | **0.000** | 14818.588 | **14720.097** |
| Qwen3-8B | **0.000** | 15522.392 | **15425.101** |
| gemini-2.5-flash | 0.077 | 807.553 | 806.420 |
| gemini-2.5-flash-lite | **0.000** | 1574.306 | **1555.886** |
| gemini-2.5-pro | 0.201 | 642.222 | 642.583 |
| gpt-5-2025-08-07 | **0.001** | 484.186 | **474.503** |
| gpt-5-mini-2025-08-07 | **0.000** | 1044.689 | **1023.604** |
| gpt-5-nano-2025-08-07 | **0.000** | 1520.921 | **1500.138** |
| o3-2025-04-16 | 0.121 | 692.757 | 692.356 |
| o4-mini-2025-04-16 | **0.000** | 1002.927 | **970.994** |

Table 6: Model comparison: linear vs. quadratic fit. A bold number indicates a significant improvement of $AIC_{quad}$ over $AIC_{linear}$ with $p < 0.05$.

Victor, Jesse, Zachary, Kyle, Aaron, Betty, Connie, Holly, Donna, Gloria, Carla, Isabel, Sylvia, Evelyn, Doris, Arthur, Raymond, Harold, Lawrence, Neil, Brenda, Tracy, Simon, Wendy, Zoe, Ethan, Calvin, Sean, Ruth, Sheila, Miriam, Lorraine, Fay, Sophie

**clothes_socks**   blue, red, yellow, green, purple, pink, orange, black, white, gray

**clothes_shirt**   blue, red, yellow, green, purple, pink, orange, black, white, gray

**clothes_pant**   blue, red, yellow, green, purple, pink, orange, black, white, gray

**clothes_hat**   blue, red, yellow, green, purple, pink, orange, black, white, gray

**clothes_gloves**   blue, red, yellow, green, purple, pink, orange, black, white, gray

**clothes_underwear**   blue, red, yellow, green, purple, pink, orange, black, white, gray

**hair**   blue, red, yellow, green, purple, pink, orange, black, white, gray

**recent_eat**   pizza pasta burrito sushi taco burger toast egg banana potatoes

**recent_listen**   rock, pop, country, electronic, folk, jazz, blues, classical, funk, ska, rap, synth, disco, reaggea

**recent_watch**   drama, comedy, thriller, romance, adventure, horror, sci-fi, action, western, fantasy, documentary, mystery, crime, musical

**recent_read**   fiction, mystery, novel, thriller, biography, sci-fi, non-fiction, essay, encyclopedia, dictionary

**location**   bathroom livingroom kitchen basement toilet balcony garden pool bedroom store university farm office bank tree museum school airport zoo train bus park butcher library restaurant mall mountain tunnel church river pond harbor taxi gallery bar pizzeria beach gym elevator insurance embassy police hospital festival monument laboratory observatory valley motorway viewpoint synagogue factory castle cave stadium arena cabin plaza amphitheater bridge pier vineyard forest cliff desert creek bay lighthouse orchard resort camp inn motel aquarium bazaar chapel monastery lookout campground retreat dock depot consulate manor theatre cathedral casino lodge mill bakery spa station diner gazebo terrace arcade boardwalk winery hill plateau ridge port oasis market fairground quarry mine grove auditorium cemetery dunes courthouse prison fort granary ranch promenade coliseum field tower pavilion silo bistro labyrinth cafe saloon brewery carnival marina estate safari cottage courtyard waterpark island greenhouse meadow lagoon ford hacienda village marketplace grotto maze golfcourse atrium academy waterfront peninsula cove summit plains

## H   Token-length distributions by task length (N) and difficulty (d)

We report token-length distributions per model tokenizer across the full CogniLoad grid in Table 7, Table 8, Table 9, and Table 10. For each model and each $(N, d)$ condition, we show the mean and the 90th percentile (p90) of both the input (prompt) tokens and the output (reasoning) tokens, compactly formatted as Pmean[Pp90] | Omean[Op90]. For example, a cell like P3995[4683] | O14778[25212] indicates that the mean prompt length is $\approx 3995$ tokens with p90 $\approx 4683$, and the mean output length is about $\approx 14778$ tokens with p90 $\approx 25212$ for that condition. Presenting both prompt and output side by side makes it possible to separate the load imposed by the puzzle itself from model and tokenizer-specific verbosity in decoding.

Across all models, prompt length grows monotonically with N and increases with d, as expected from the construction of the puzzles. At fixed $(N, d)$, the variation in prompt tokens across tokenizers is small (i.e. typically a few percent) which indicates that models received essentially the same input burden for a given condition. For example, at $N = 250$ and $d = 10$ as the highest load condition, prompt means cluster around 16000–17000 tokens, while at $d = 1$ they are around 3500–4000 with these patterns recurring for every tokenizer. This confirms that differences in performance at fixed $(N, d)$ do not stem from large differences in presented input length.

On the contrary, output lengths vary substantially across models and show heavy-tailed behavior in several cases, especially at larger N. Gemini 2.5 models tend to produce the longest outputs: at $N = 250$ and moderate d, their output p90 values are frequently in the mid-20000s while prompt p90 values are often around 6500–7000, so totals approach or exceed the 32768-token budget. For instance, at $N = 250$ and $d = 5$, gemini-2.5-pro shows approximately P9474[10084] | O23221[24321], and gemini-2.5-flash is in a similar range. These totals explain the higher rates of max-context overflows we observed for Gemini at high N indicating a serving limitation rather than genuine reasoning error. By comparison, o3 and the gpt-5 family are more restrained in their output lengths at the same $(N, d)$: their output p90s are typically around 11000–19000 at $N = 250$ depending on d, leaving more headroom when paired with prompt p90s near 6500–7000. Mid-tier open models such as DeepSeek-R1-Distill-Llama-70B, Qwen3-32B, QwQ-32B, and EXAONE-Deep-32B generally produce shorter outputs still, for example, Qwen3-32B at $N = 250$ and $d = 7$ shows roughly P12419[13055] | O5034[8153], a total far below the context limit. Smaller models often have shorter outputs on average but can occasionally exhibit very long tails. Inspecting their transparent reasoning traces we notice those outliers being consistent with the error taxonomy's degeneracy categories (e.g., wrong-max-context or "other"), where models infinitely repeat tokens or drift into infinite reasoning loops rather than concluding. One peculiar case is Gemini-2.5-flash-lite at the cell $N{=}250/d{=}10$ resulting in P16683[17372] | O366[9]. Here this particular model doesn't even try to reason through the problem but usually just guesses the response with barely any reasoning.

Taken together, these distributions show that prompt length scales predictably with N and d and is closely matched across tokenizers, while output length is where models differ. In our runs, most models and conditions—including many at $N = 250$—remain well below 32768 tokens in total, indicating that CogniLoad puzzles are generally solvable within a 32K context. Overflows are concentrated in specific settings where certain models (notably Gemini 2.5 at high N) produce heavy-tailed outputs while smaller models show occasional outliers that, when inspecting the reasoning traces,

boil down to infinite reasoning loops and incoherence issues that repeat the same tokens indefinitely. Consequently, for the majority of models and conditions the observed performance trends at long N reflect reasoning under load rather than system ceilings.

| Model | d | | | | |
| --- | --- | --- | --- | --- | --- |
| | 1 | 3 | 5 | 7 | 10 |
| DeepSeek-R1-0528 | P400[424]—O4135[6756] | P626[690]—O4697[5531] | P924[1008]—O6194[7477] | P1218[1358]—O6901[8474] | P1728[1958]—O8225[9710] |
| DeepSeek-R1-Distill-Llama-70B | P358[404]—O1686[2739] | P616[683]—O2361[3245] | P904[996]—O2456[3364] | P1209[1348]—O2829[3972] | P1681[1924]—O2473[4067] |
| DeepSeek-R1-Distill-Qwen-1.5B | P369[415]—O23722[119748] | P627[694]—O22617[64932] | P915[1007]—O20799[61653] | P1220[1359]—O17447[37955] | P1692[1935]—O17871[36281] |
| DeepSeek-R1-Distill-Qwen-32B | P369[415]—O2447[3914] | P627[694]—O2891[4668] | P915[1007]—O2686[3569] | P1220[1359]—O2776[3908] | P1692[1935]—O2941[4668] |
| DeepSeek-R1-Distill-Qwen-7B | P369[415]—O3492[5413] | P627[694]—O4771[6554] | P915[1007]—O5635[7088] | P1220[1359]—O5842[8063] | P1692[1935]—O4979[6996] |
| EXAONE-Deep-32B | P393[438]—O7086[11418] | P653[721]—O9284[12422] | P945[1040]—O11174[14357] | P1251[1393]—O12002[15366] | P1730[1979]—O10971[15880] |
| Phi-4-mini-reasoning | P369[416]—O5944[9174] | P625[690]—O8012[10967] | P913[1003]—O9423[12726] | P1216[1353]—O9941[13258] | P1687[1929]—O9417[12983] |
| Phi-4-reasoning | P579[625]—O3663[5530] | P837[904]—O5182[6575] | P1125[1217]—O6682[8409] | P1430[1569]—O7971[10066] | P1902[2145]—O9835[12502] |
| Phi-4-reasoning-plus | P556[625]—O5342[8017] | P823[896]—O6319[8146] | P1124[1218]—O8266[10849] | P1442[1597]—O10442[13677] | P1932[2206]—O13151[17120] |
| QwQ-32B | P374[420]—O4194[7729] | P632[699]—O4699[6743] | P920[1012]—O5032[6622] | P1225[1364]—O5239[7108] | P1697[1940]—O5399[7909] |
| Qwen3-1.7B | P372[418]—O4774[7658] | P630[697]—O5857[7752] | P918[1010]—O4129[6464] | P1223[1362]—O3030[5337] | P1695[1938]—O2067[4015] |
| Qwen3-30B-A3B | P372[418]—O3030[5608] | P630[697]—O4673[6464] | P918[1010]—O4107[5337] | P1223[1362]—O3546[4907] | P1695[1938]—O3141[5082] |
| Qwen3-32B | P372[418]—O2307[4211] | P630[697]—O2935[4758] | P918[1010]—O3054[4355] | P1223[1362]—O2902[4322] | P1695[1938]—O2928[4502] |
| Qwen3-8B | P372[418]—O4262[6898] | P630[697]—O5222[6698] | P918[1010]—O5048[6469] | P1223[1362]—O4789[6324] | P1695[1938]—O3967[5742] |
| gemini-2.5-flash | P364[411]—O2130[2924] | P625[696]—O3947[5396] | P900[997]—O6015[8151] | P1217[1345]—O6926[8780] | P1699[1935]—O7769[10955] |
| gemini-2.5-flash-lite | P364[411]—O1073[1246] | P625[696]—O1700[2510] | P900[997]—O1897[2850] | P1217[1345]—O2002[3016] | P1699[1935]—O1851[3585] |
| gemini-2.5-pro | P364[411]—O2784[3842] | P625[696]—O4398[6015] | P900[997]—O5895[7953] | P1217[1345]—O7360[9669] | P1699[1935]—O8311[11803] |
| gpt-5-2025-08-07 | P357[404]—O1228[1750] | P615[685]—O2768[3534] | P889[984]—O3449[4250] | P1207[1334]—O3682[4627] | P1686[1914]—O4256[5212] |
| gpt-5-mini-2025-08-07 | P357[404]—O1670[2313] | P615[685]—O3066[3555] | P889[984]—O4256[5302] | P1215[1351]—O5182[6320] | P1686[1914]—O6277[7873] |
| gpt-5-nano-2025-08-07 | P357[404]—O2633[5018] | P615[685]—O4444[6470] | P889[984]—O5917[9075] | P1207[1334]—O6704[9436] | P1686[1914]—O8127[12956] |
| o3-2025-04-16 | P357[404]—O933[1359] | P615[685]—O2126[2580] | P889[984]—O2685[3213] | P1207[1334]—O2933[3866] | P1686[1914]—O3557[4685] |
| o4-mini-2025-04-16 | P357[404]—O1682[2462] | P615[685]—O2743[3545] | P889[984]—O3352[4253] | P1207[1334]—O3686[4569] | P1686[1914]—O4503[5861] |

Table 7: Prompt and output tokens by model at N=20. Cells show Pmean[Pp]—Omean[Op], Pp=Op=p90.

| Model | d | | | | |
| --- | --- | --- | --- | --- | --- |
| | 1 | 3 | 5 | 7 | 10 |
| DeepSeek-R1-0528 | P779[908]—O6060[8073] | P1372[1537]—O7714[9286] | P2021[2195]—O10932[13181] | P2682[2900]—O12144[15058] | P3658[3946]—O12701[15868] |
| DeepSeek-R1-Distill-Llama-70B | P774[884]—O2503[3688] | P1362[1500]—O4136[5655] | P1987[2143]—O4427[6201] | P2652[2851]—O4129[6600] | P3669[3933]—O3245[5604] |
| DeepSeek-R1-Distill-Qwen-1.5B | P815[925]—O17222[36431] | P1403[1541]—O14514[33405] | P2028[2184]—O13125[32676] | P2693[2892]—O12702[32454] | P3710[3974]—O14078[32989] |
| DeepSeek-R1-Distill-Qwen-32B | P815[925]—O3305[5198] | P1403[1541]—O4753[6624] | P2028[2184]—O467[6376] | P2693[2892]—O4820[7171] | P3710[3974]—O4162[7217] |
| DeepSeek-R1-Distill-Qwen-7B | P815[925]—O4721[7396] | P1403[1541]—O6534[9242] | P2028[2184]—O7725[11797] | P2693[2892]—O8318[29548] | P3710[3974]—O5995[26866] |
| EXAONE-Deep-32B | P842[950]—O10773[14601] | P1432[1569]—O13349[16687] | P2064[2221]—O14859[19210] | P2737[2937]—O12815[17829] | P3758[4027]—O9117[13680] |
| Phi-4-mini-reasoning | P784[896]—O7723[10840] | P1369[1497]—O9218[12190] | P1992[2145]—O9301[13126] | P2654[2848]—O8619[12060] | P3667[3929]—O7220[10087] |
| Phi-4-reasoning | P995[1105]—O5733[8056] | P1583[1721]—O8164[10198] | P2208[2364]—O11462[14776] | P2874[3073]—O13659[19401] | P3890[4154]—O15917[28612] |
| Phi-4-reasoning-plus | P983[1105]—O7885[11340] | P1596[1745]—O10537[13481] | P2240[2419]—O14875[19947] | P2931[3218]—O19655[29629] | P4006[4619]—O21226[28904] |
| QwQ-32B | P820[930]—O5988[9091] | P1408[1546]—O6617[8934] | P2033[2189]—O7580[10243] | P2698[2897]—O7937[11646] | P3715[3979]—O6928[11473] |
| Qwen3-1.7B | P818[928]—O4180[7015] | P1406[1544]—O5482[7856] | P2031[2187]—O3682[6167] | P2696[2895]—O2492[4879] | P3713[3977]—O1867[4431] |
| Qwen3-30B-A3B | P818[928]—O3847[5532] | P1406[1544]—O5600[7227] | P2031[2187]—O5240[6894] | P2696[2895]—O5087[7280] | P3713[3977]—O4451[7365] |
| Qwen3-32B | P818[928]—O3547[5774] | P1406[1544]—O4416[6680] | P2031[2187]—O4189[6276] | P2696[2895]—O4045[6100] | P3713[3977]—O3730[6143] |
| Qwen3-8B | P818[928]—O5183[7561] | P1406[1544]—O6971[8185] | P2031[2187]—O6023[8164] | P2696[2895]—O5784[7660] | P3713[3977]—O4879[7485] |
| gemini-2.5-flash | P810[922]—O4343[6115] | P1373[1501]—O7708[10836] | P2030[2208]—O12945[17344] | P2688[2883]—O16780[22526] | P3691[3957]—O18449[28927] |
| gemini-2.5-flash-lite | P810[922]—O1840[2739] | P1373[1501]—O2971[5314] | P2030[2208]—O3570[5976] | P2688[2883]—O2867[5504] | P3691[3957]—O1510[5970] |
| gemini-2.5-pro | P810[922]—O5958[8294] | P1373[1501]—O9435[13393] | P2030[2208]—O12585[17418] | P2688[2883]—O14723[20703] | P3691[3957]—O16236[21429] |
| gpt-5-2025-08-07 | P770[884]—O2112[2900] | P1335[1463]—O5251[6805] | P1992[2169]—O5976[7904] | P2644[2830]—O6239[8467] | P3646[3914]—O6534[8732] |
| gpt-5-mini-2025-08-07 | P770[884]—O2670[3788] | P1335[1463]—O5794[6779] | P1992[2169]—O8959[11716] | P2644[2830]—O10562[14197] | P3646[3914]—O9879[15575] |
| gpt-5-nano-2025-08-07 | P770[884]—O4146[6810] | P1335[1463]—O7488[11220] | P1992[2169]—O10491[15707] | P2914[2878]—O10557[16212] | P3646[3914]—O9268[16136] |
| o3-2025-04-16 | P770[884]—O1704[2388] | P1335[1463]—O4123[5006] | P1992[2169]—O5237[7181] | P2644[2830]—O5371[8103] | P3646[3914]—O5411[7861] |
| o4-mini-2025-04-16 | P770[884]—O2692[4426] | P1335[1463]—O4439[5671] | P1992[2169]—O6120[9511] | P2644[2830]—O5833[9118] | P3646[3914]—O6446[9573] |

Table 8: Prompt and output tokens by model at N=50. Cells show Pmean[Pp]—Omean[Op], Pp=Op=p90.

| Model | 1 | 3 | d 5 | 7 | 10 |
|---|---|---|---|---|---|
| DeepSeek-R1-0528 | P1457[1684]—O8142[11134] | P2627[2930]—O12136[14418] | P3873[4119]—O14319[18636] | P5035[5376]—O11861[17026] | P6860[7253]—O10608[13931] |
| DeepSeek-R1-Distill-Llama-70B | P1468[1684]—O3628[5573] | P2630[2904]—O6596[9646] | P3793[4071]—O6587[10134] | P5009[5308]—O5706[10984] | P6878[7239]—O4133[8443] |
| DeepSeek-R1-Distill-Qwen-1.5B | P1560[1776]—O16989[34188] | P2722[2996]—O12228[32306] | P3885[4163]—O10608[31366] | P5101[5400]—O10769[29845] | P6970[7331]—O9680[28165] |
| DeepSeek-R1-Distill-Qwen-32B | P1560[1776]—O4480[7023] | P2722[2996]—O7847[11083] | P3885[4163]—O7881[11277] | P5101[5400]—O7047[11282] | P6970[7331]—O5578[10228] |
| DeepSeek-R1-Distill-Qwen-7B | P1560[1776]—O5504[9157] | P2722[2996]—O8825[14593] | P3885[4163]—O10153[32087] | P5101[5400]—O8469[31107] | P6970[7331]—O8149[30381] |
| EXAONE-Deep-32B | P1588[1859]—O13168[17829] | P2758[3051]—O15139[18752] | P3931[4219]—O13399[18030] | P5160[5470]—O10195[14815] | P7034[7399]—O7605[11342] |
| Phi-4-mini-reasoning | P1475[1695]—O8778[12190] | P2633[2908]—O8882[11839] | P3792[4060]—O8457[12413] | P5003[5302]—O7324[10152] | P6864[7220]—O6699[9230] |
| Phi-4-reasoning | P1689[1905]—O8360[11469] | P2851[3125]—O12068[15496] | P4015[4295]—O15167[27514] | P5230[5529]—O16521[27514] | P7103[7462]—O16996[25674] |
| Phi-4-reasoning-plus | P1694[1968]—O12270[16909] | P2910[3238]—O15333[21025] | P4123[4586]—O20288[28560] | P5392[6208]—O19976[27662] | P7309[8426]—O19392[25782] |
| QwQ-32B | P1565[1781]—O7625[10956] | P2727[3001]—O9009[11892] | P3890[4168]—O9089[13166] | P5106[5405]—O9096[13950] | P6975[7336]—O8298[12846] |
| Qwen3-1.7B | P1563[1779]—O4079[7309] | P2725[2999]—O5224[9101] | P3888[4166]—O4191[7665] | P5104[5403]—O2933[6423] | P6973[7334]—O2144[7498] |
| Qwen3-30B-A3B | P1563[1779]—O4949[7052] | P2725[2999]—O6596[8505] | P3888[4166]—O6443[9249] | P5104[5403]—O5233[8618] | P6973[7334]—O4263[8469] |
| Qwen3-32B | P1563[1779]—O5141[7983] | P2725[2999]—O5631[8177] | P3888[4166]—O5157[7311] | P5104[5403]—O4638[7052] | P6973[7334]—O3961[6269] |
| Qwen3-8B | P1563[1779]—O6076[8098] | P2725[2999]—O7797[10322] | P3888[4166]—O7659[10472] | P5104[5403]—O7076[10829] | P6973[7334]—O5596[10241] |
| gemini-2.5-flash | P1536[1771]—O8290[13046] | P2699[2990]—O14984[20409] | P3910[4123]—O25507[29050] | P5087[5439]—O25224[28010] | P6909[7279]—O23637[26097] |
| gemini-2.5-flash-lite | P1536[1771]—O2526[3917] | P2699[2990]—O5072[8372] | P3910[4123]—O5391[10345] | P5087[5439]—O2846[8615] | P6909[7279]—O16121[7689] |
| gemini-2.5-pro | P1536[1771]—O10927[15454] | P2699[2990]—O16454[21166] | P3910[4123]—O21258[28842] | P5087[5439]—O21546[27761] | P6909[7279]—O22577[26042] |
| gpt-5-2025-08-07 | P1446[1683]—O3294[4693] | P2606[2884]—O7840[10062] | P3814[4016]—O8165[11989] | P4992[5333]—O8733[13057] | P6808[7173]—O9283[12365] |
| gpt-5-mini-2025-08-07 | P1446[1683]—O3808[5621] | P2606[2884]—O9745[11827] | P3814[4016]—O13104[18082] | P5037[5337]—O12790[21517] | P6808[7173]—O12440[17742] |
| gpt-5-nano-2025-08-07 | P1607[1683]—O5999[9006] | P2627[2905]—O10037[15855] | P3814[4016]—O13450[19789] | P4992[5333]—O9336[17104] | P6808[7173]—O8322[15122] |
| o3-2025-04-16 | P1446[1683]—O2717[4066] | P2606[2884]—O7151[9366] | P3814[4016]—O6959[10925] | P4992[5333]—O6897[10914] | P6808[7173]—O8748[14425] |
| o4-mini-2025-04-16 | P1446[1683]—O4153[6167] | P2606[2884]—O6444[9652] | P3814[4016]—O7910[13477] | P4992[5333]—O7269[12320] | P6808[7173]—O8365[16052] |

Table 9: Prompt and output tokens by model at N=100. Cells show Pmean[Pp]—Omean[Op], Pp=Op=p90.

| Model | 1 | 3 | 5 (d) | 7 | 10 |
|---|---|---|---|---|---|
| DeepSeek-R1-0528 | P3639[4255]—O11660[15547] | P6442[6992]—O12139[15672] | P9132[9696]—O10608[13547] | P12014[12793]—O10417[13038] | P16345[17021]—O9524[11707] |
| DeepSeek-R1-Distill-Llama-70B | P3547[4084]—O4738[7376] | P6355[6974]—O10638[18791] | P9173[9789]—O8206[16317] | P12024[12660]—O6814[15722] | P16395[16998]—O4547[11339] |
| DeepSeek-R1-Distill-Qwen-1.5B | P3939[4476]—O16892[32864] | P6747[7366]—O10229[28441] | P9565[10181]—O11895[28483] | P12416[13052]—O11754[28378] | P16787[17390]—O8652[19503] |
| DeepSeek-R1-Distill-Qwen-32B | P3939[4476]—O5778[10068] | P6747[7366]—O16565[23979] | P9565[10181]—O15297[23500] | P12416[13052]—O12679[22803] | P16787[17390]—O9262[21900] |
| DeepSeek-R1-Distill-Qwen-7B | P3977[4500]—O12691[16997] | P6801[7452]—O12493[17556] | P9641[10282]—O11156[15514] | P12513[13177]—O8395[12329] | P16787[17390]—O7354[23761] |
| EXAONE-Deep-32B | P3550[4096]—O8854[12622] | P6345[6934]—O7673[10696] | P9153[9758]—O7091[10159] | P11993[12610]—O6053[8649] | P16933[17557]—O5523[9281] |
| Phi-4-mini-reasoning | P3768[4305]—O14056[19926] | P6576[7195]—O17766[25623] | P9394[10010]—O18249[23605] | P12245[12881]—O18222[20994] | P16345[16948]—O5462[7771] |
| Phi-4-reasoning | P3830[4315]—O20039[28464] | P6796[7802]—O20861[26050] | P9739[11372]—O20595[23689] | P12670[14810]—O19383[21013] | P16616[17219]—O14578[16740] |
| Phi-4-reasoning-plus | P3944[4481]—O9353[13906] | P6752[7371]—O14473[19810] | P9570[10186]—O13878[21556] | P12421[13057]—O12844[20036] | P17347[20890]—O15076[16718] |
| QwQ-32B | P3942[4479]—O3919[7534] | P6750[7369]—O6392[14932] | P9568[10184]—O5147[12851] | P12419[13055]—O4673[12036] | P16792[17395]—O10085[16212] |
| Qwen3-1.7B | P3942[4479]—O7036[10502] | P6750[7369]—O8278[14676] | P9568[10184]—O7500[14503] | P12419[13055]—O5664[13342] | P16792[17394]—O2718[13976] |
| Qwen3-30B-A3B | P3942[4479]—O6840[9827] | P6750[7369]—O7084[10285] | P9568[10184]—O6474[9809] | P12419[13055]—O5034[8153] | P16790[17393]—O3418[8000] |
| Qwen3-32B | P3942[4479]—O8146[12378] | P6750[7369]—O10552[20393] | P9568[10184]—O9710[18564] | P12419[13055]—O8019[17663] | P16790[17393]—O3654[6255] |
| Qwen3-8B | P3995[4683]—O14778[25212] | P6768[7256]—O26171[26820] | P9474[10084]—O23697[24363] | P12351[13131]—O20829[21494] | P16790[17393]—O5349[13378] |
| gemini-2.5-flash | P3995[4683]—O5843[10440] | P6768[7256]—O11447[19464] | P9474[10084]—O6728[21344] | P12351[13131]—O3665[19711] | P16683[17372]—O16508[17088] |
| gemini-2.5-flash-lite | P3995[4683]—O21432[28925] | P6768[7256]—O26299[26895] | P9474[10084]—O23221[24321] | P12351[13131]—O20527[21494] | P16683[17372]—O366[9] |
| gemini-2.5-pro | P3596[4232]—O5910[7532] | P6376[6874]—O11577[16352] | P9076[9672]—O13513[17813] | P11944[12654]—O14327[19091] | P16683[17372]—O16395[17073] |
| gpt-5-2025-08-07 | P3596[4232]—O6472[8818] | P6376[6874]—O15706[23134] | P9076[9672]—O15604[20850] | P11944[12654]—O15981[20713] | P16265[16934]—O13799[16902] |
| gpt-5-mini-2025-08-07 | P3762[4246]—O10612[16059] | P6376[6874]—O8616[17818] | P9076[9672]—O8662[15925] | P11944[12654]—O6539[16148] | P16274[16934]—O13598[16832] |
| gpt-5-nano-2025-08-07 | P3589[4232]—O4971[7131] | P6376[6874]—O11299[15104] | P9074[9673]—O10748[16757] | P11944[12654]—O12022[19860] | P16267[16934]—O4498[11784] |
| o3-2025-04-16 | P3604[4232]—O6679[9945] | P6376[6874]—O10127[16494] | P9076[9672]—O11155[19673] | P11948[12662]—O10845[19674] | P16265[16934]—O11833[16704] |
| o4-mini-2025-04-16 | P3604[4232]—O6679[9945] | P6376[6874]—O10127[16494] | P9076[9672]—O11155[19673] | P11948[12662]—O10845[19674] | P16265[16934]—O10263[16382] |

Table 10: Prompt and output tokens by model at N=250. Cells show Pmean[Pp]—Omean[Op], Pp=Op=p90.

