# OpenReview forum: "CogniLoad: A Synthetic Natural Language Reasoning Benchmark With Tunable Length, Intrinsic Difficulty, and Distractor Density"
_ICLR.cc/2026/Conference — ICLR 2026 Poster_

### Official Review · Reviewer_aNzL · 2025-10-28

**Soundness:** 4
**Presentation:** 3
**Contribution:** 2
**Rating:** 6
**Confidence:** 3

**Summary:**

The paper introduces a benchmark for long-context reasoning problems based on Cognitive Load Theory, which categorizes cognitive demands into three categories: intrinsic, extrinsic, and germane. Using this framework, they create puzzles in deductive reasoning and state tracking. They describe a generation procedure to change the difficulty of the benchmark by changing each of these dimensions separately. They generate 14,000 (4x5x7x100) puzzles to evaluate 22 LLMs across the dimensions. They fit a GLM and show the performance of the LLMs with respect to variations in each of the three dimensions, with task length still dominating.

**Strengths:**

**Originality:** This paper offers a new lens for dividing up the factors which contribute to LLM performance using cognitive loading.

**Quality:** The results in the paper are sound and thorough. I am particularly pleased that the authors used statistical methods to perform their analysis of benchmark results, and have a large sample size.

**Clarity:** The description of the approach is described in detailed mathematical notation, making it highly precise.

**Significance:** Meaningful dimensions for the analysis of LLM results are important to guide research, and I find comparisons to cognitive science to be particularly interesting, as they draw on well-established concepts (though of course with the caveat that LLMs do not have human minds...). The algorithm for procedurally-generated benchmark items is also useful.

**Weaknesses:**

I find the paper to be well-executed, and have minimal concerns about the quality of the results. The concerns that I do have come from unclear or missing details in the paper, grounding in cognitive science, and from uncertainty about the broader impact of the work.

**Soundness:** As a sense check, I do want to know that the task is possible within the context lengths provided. As with another recent paper (https://arxiv.org/abs/2506.06941) models running over the context window seems like a poor justification for calling a performance a "reasoning" failure.

**Clarity:**

1. The mathematical notation of the paper was comprehensive, but not always helpful. I think some pieces were missing, such as the use of k_t and m_t after their initial definition on line 199-200. I would also suggest distinguishing between "=" as a logical operator and "=" as a definition (probably by using ":="). v_{c,t} and u_{c,t} were a bit confusing along this lines, when moving between the definitions and their use validating statements.

2. I am not certain how the 14000 problems were actually finally generated. Was this done with an LLM? Did you create formal logical sets and then translate them into plain language? The latter certainly seems more reliable.

(I will note that although there were references to supplementary material, I only found a zip file with code and data, not further text. In any case, I do think that these details should be in the main body of the paper.)

**Cognitive Science:** I am not a cognitive scientist, but I felt that the grounding in this theory was relatively light. I would appreciate more explanation/justification about why d, N, \rho are the right way to operationalize these dimensions, beyond what appears in Table 1. In particular, my intuition is that d does not capture "difficulty" in the way I would usually conceive of it, since higher values require more working memory, but not more complex deductions. Similarly, I would imagine that instruction following is a subset of ECL, but seems to be described as outside of the three dimensions in the error analysis. One way to make this argument would be by comparison to a human exam that varies along the same cognitive dimensions, to show an analogy between how those variations are made and the variations made in this puzzle.

**Significance:** I am open to being wrong here, but I worry that this paper does not show us things we didn't already know. State tracking is a known weakness of LLMs (https://arxiv.org/abs/2404.08819, https://proceedings.neurips.cc/paper_files/paper/2024/hash/e6231c5f46598cfd09ff1970524e0436-Abstract-Conference.html). It is also a relatively simple task to be testing relative to the more complex reasoning benchmarks described in related works. The variation of performance along context length (most previous works), task size (https://arxiv.org/abs/2506.06941) and number of needles (https://arxiv.org/pdf/2404.06654) have also been shown previously. Being able to track these three dimensions separately is probably new, but I would like to see a clearer discussion of why this matters.

I also find the error analysis to be somewhat unhelpful. Finding that the models often run out of context is described as as failure to use tokens efficiently, but efficient token use is not something that the benchmark describes itself as measuring. Why shouldn't the task be adjusted to avoid these issues in order to ensure a pure assessment of the target reasoning abilities?

Incidentally, I think addressing this can be tied to the previous comment about grounding in cognitive science. To the extent that the dimensions are well-justified, papers like (https://arxiv.org/html/2503.06378v2) have shown useful predictive results from cognitive science dimensions.

**Overall**: I have netted out to a weak accept because I believe the work is methodologically sound, and want to be charitable about the significance, but I am very open to having my opinion swayed, particularly in how to weight the different elements of the papers.

**Questions:**

1. What is the distribution of lengths of the puzzles in tokens? What fraction of the context window is consumed by the inputs? This task seems to require tracking a considerable amount of information. How much context would actually be required to just write out the states after each step?

2. What do the findings of the error analysis mean for people trying to improve LLMs? Would training models via RL to perform state tracking tasks using the context window be a valuable thing to do?

---

> ### Author Response · Authors · 2025-11-18
>
> ## Soundness: As a sense check, I do want to know that the task is possible within the context lengths provided.
>
> The updated appendix (section H) now reports, for every model and each (N, d) cell, the mean and 90th percentile of both prompt and output token counts. The data show that the bottleneck is the output length as some models (i.e. Gemini-2.5) generate substantially longer reasoning traces than all others. When added to the shared prompts, these long generations are what push totals to the 32 768‑token window.  However for the majority of models and conditions the mean total (prompt + output) remains well below the 32 768‑token budget.
>
> As mentioned, this pattern is most pronounced for the Gemini 2.5 models at N = 250, where output p90s are often in the mid‑20 000s. By contrast, o3 and the gpt‑5 family keep output p90s much lower ($\approx$ 11 000–19 000 at N = 250, depending on d), leaving substantial headroom under the same prompts. While this token usage is comfortably below the token limit, performance still drops for these models.  Mid‑tier open models typically remain lower still. This pattern demonstrates that the tasks are generally solvable within a 32 K window, and that overflows are concentrated in specific model–condition combinations rather than indicating that the benchmark generally requires more context than provided.
>
> While we highlight “max‑context overflow” as a separate error category and discuss it in the error analysis as such, you are correct that they should also not influence the main reported results in figure 1 and the GLM. We will therefore re‑run the high‑N regimes for the Gemini 2.5 models (which are primarily affected by this) with a higher context budget and update their reported results, so that their performance is not affected by this serving limit.
>
> ## Clarity: The mathematical notation of the paper was comprehensive, but not always helpful.
>
> Thank you for highlighting this. We update the mathematical notation accordingly to improve clarity.
>
> ## Not certain how the 14000 problems were actually finally generated.
>
> The link to the anonymized code repository which we provided in the submission (https://anonymous.4open.science/r/cogniload-292B/README.md)  contains the python code that generated the puzzles and their text. It is an exact implementation of the algorithm we describe in the paper and converts the logical state transitions of the puzzle into plain language using string templates. We will state this more clearly in the paper.
>
> In the supplementary materials we provided a pdf with further text (e.g. additional experiments, error analysis, attribute ontology, …) and see that it is correctly linked in the OpenReview submission (https://openreview.net/attachment?id=0Sex2H5Jnn&name=supplementary_material).
>
> ## d does not capture "difficulty" and instruction following is a subset of ECL
>
> As you correctly state, higher d does require more memory (i.e. includes more attributes per person) but also creates more complex deductions by making the deduction statements conditional on more properties as well as by assigning more value updates per statement. In the terminology of CLT, it increases element interactivity and working-memory requirements (making it an accurate measure for intrinsic load). If it is of interest, please also see  our answer to the first question of reviewer je6v where we explain germane load in more detail.
>
> The “instruction following” which we mention in the error analysis refers to following the instruction in the prompt to respond in one sentence mentioning the person and the final attribute (i.e. “You MUST finalize your response with a single sentence about the asked property (e.g., "Peter is in the livingroom.", "Peter is wearing blue socks",.. ). In this context we refer to it outside of the three CLT dimensions because it is not related to the task properties and ultimately the required reasoning to solve the task - it just refers to the instruction to format the answer in a certain way (see related question about this reviewer 2 concern 2). We thank the reviewer for making us aware that we used a confusing terminology in a paper for a benchmark that consists of following deductive instructions and therefore we will make the text clearer.

---

> > ### Author Response · Authors · 2025-11-18
> >
> > ## Significance: State tracking is a known weakness of LLMs & "simple task"
> >
> > We agree that “state tracking” is a known weakness of LLMs and that prior work has varied context length, task size, and needle density alone. Our goal is broader than state tracking per se with key novelties and why they matter being:
> > - Factorial, CLT-grounded control on one standardized task family: We operationalize intrinsic load (ICL), extraneous load (ECL), and conditions demanding germane-like processing via three independently tunable parameters within the same natural‑language puzzle semantics. In contrast, prior benchmarks typically vary a single axis or collapse multiple factors across different datasets (e.g., LongBench/L‑Eval for length; LogicBench/CLUTRR/ZebraLogic for intrinsic complexity; NIAH/RULER and related for distractors). This deconfounding is what enables precise attribution.
> > - Joint statistical analysis enables relative importance and interpretable “capacity” measures:  Because d, N, and ρ are controlled within each single task family, we can fit a single GLM per model and derive interpretable relative thresholds - ECL50 (effective context length at 50% accuracy), ID50 (intrinsic difficulty at 50%), and NT50 (minimum relevant‑information ratio for 50%). This yields actionable, model‑specific “cognitive fingerprints” that are impossible to obtain when axes are probed on different datasets with different tasks. Also the interaction analysis of the three variables which is included in the updated Appendix, would be impossible to conduct without sharing these three dimensions across a single task.
> > - Automatic randomized generation that prevents contamination and memorization.   Our generator produces arbitrarily many fresh puzzle instances with clear gold states (i.e. state transitions, used attributes) in the metadata, mitigating test‑set leakage and memorization concerns and thereby adding to the longevity of the benchmark.
> >
> >
> > Tracking the three dimensions separate dimensions matters because of:
> > - Attribution: Without factorial control, the same accuracy drop could be due to length, intrinsic interactivity, or distractors. Our design isolates each, allowing clean main‑effect estimates.
> > - Comparability: Prior works probe length, difficulty, or distractors on different corpora, tasks and instructions, making it impossible to accurately evaluate effect sizes. We measure all three on identical semantics and instructions, so effect magnitudes are directly comparable.
> > - Interactions and trade‑offs: The joint design reveals effects that single‑axis tests miss. For example, we observe a robust U‑shaped response to \rho (worst at moderate clutter) and that increases in N flatten $\rho$‑sensitivity, indicating a shift to length‑dominated failure. This kind of interaction is invisible when varying only needles or only length on separate datasets.
> > - Actionability: Capacity thresholds (ECL50, ID50, NT50) and sensitivity profiles identify where to invest: e.g. length‑robust attention/memory improves ECL50, while better selective filtering helps NT50. This goes beyond stating that “state tracking is hard” by pointing towards where and which feature of the task makes it hard.
> >
> >
> > While we acknowledge the literature showing that LLMs struggle with long contexts, distractors, and complex logic, CogniLoad’s contribution is to provide a single, CLT‑grounded, factorial framework that (i) cleanly attributes failure modes, (ii) quantifies relative importance via interpretable and comparable thresholds, and (iii) scales with automatic, contamination‑resistant generation.
> >
> > CogniLoad advances the field by moving from qualitative observation to quantitative attribution. By decoupling the variables, we provide actionable 'cognitive fingerprints' that allow researchers to really determine why a specific model fails. This level of diagnostic precision is impossible with existing benchmarks that confound these variables across fundamentally different tasks.
> >
> >
> > On the “simple task” concern:
> >
> > The puzzles intentionally don’t require knowledge for solving to not confound knowledge and pure reasoning ability, but they are not trivial: the model must execute a long sequence of conditional, compositional state updates, while ignoring equally plausible distractor updates. This taxes selective attention and stateful reasoning rather than retrieval of learned facts. Empirically, several strong models still collapse with N, and most exhibit the non‑obvious U‑shape in $\rho$. These behaviors are not explained by basic retrieval or simple state matching and motivate the CLT‑grounded decomposition.

---

> > > ### Author Response · Authors · 2025-11-18
> > >
> > > ## Error analysis somewhat unhelpful.
> > >
> > > The models running out of context is a rare phenomenon that occurs for very few model families (i.e. Phi-4 and Gemini 2.5) for different reasons. While for Gemini 2.5 it results from a generally higher token verbosity, that we also also observe in N<250 in responses (that fit within the limit), in the case of the Phi-4 models we see in the transparent reasoning traces that the max-context errors occur primarily because of infinite reasoning loops and models incoherently repeating a sequence of tokens indefinitely - these cases for us represent a genuine failure to solve the task. For the frontier GPT-5 and O3 models, the token usage is far below the limit, yet performance still drops proving it is a reasoning bottleneck, not a context window bottleneck.
> > >
> > > As you correctly state however, efficient token use is not something the benchmark evaluates, and for this reason we will rerun the N=250 experiments for the Gemini 2.5 models with a higher max-token limit. This limit was initially established based on pilot experiments across a wide range of models where they generally solved the task within these bounds. Only the subsequent full-scale evaluation revealed Gemini’s outlier behaviour regarding output token verbosity.
> > >
> > > ## Grounding in cognitive science and the General Scales paper
> > >
> > > CogniLoad is grounded in Cognitive Load Theory (CLT), which gives mechanistic constructs (i.e. intrinsic, extraneous, and germane load) and we causally manipulate each inside one task family via d, ρ, and N. This matters because it removes annotation noise, heterogeneity in task semantics, and resulting cross-benchmark confounds, turning “what made this task hard?” into a clean causal attribution of element interactivity (d), selective filtering under indistinguishable distractors (ρ), or sustained processing demands (N).
> > >
> > > By contrast, “General Scales” [1] achieves breadth and predictive power by annotating many benchmarks along a wide set of dimensions, some of which are task or dataset properties (e.g., Volume, Unguessability, Atypicality) rather than cognitive mechanisms per se. That breadth is valuable for coverage but it is not tied to a unified explanatory theory of cognitive load as established in cognitive psychology.
> > >
> > > Because CogniLoad is CLT‑grounded, the mechanism‑to‑prediction link is tighter and more actionable: the theory predicts (and our results confirm) how difficulty changes as each load varies (including the non‑monotonic impact of extraneous load at mid‑density distractors), and it yields clear levers for intervention. This makes CogniLoad for the specific question of cognitive load in long‑context reasoning, not only complementary to “General Scales” [1] but better founded and directly actionable through a theory‑driven, causally controlled stress test rather than an annotated aggregation.
> > >
> > > [1] https://arxiv.org/html/2503.06378v2
> > >
> > > ## Question: What is the distribution of lengths of the puzzles in tokens?
> > >
> > > We respond to this concern in our rebuttal to weakness 1 and updated the appendix (section F) to report these figures per model, d, and N.
> > >
> > > ## Question: Meaning for improving LLMs and RL in this context
> > >
> > > The error analysis indicates that most failures are not simple retrieval mistakes but sequential state-tracking breakdowns that grow with task length, with additional sensitivity to intermediate distractor densities (U-shaped effect) and, for smaller models, instruction drift. This points to improving long-term state maintenance, noise-robust attention, and token-efficiency as primary levers.
> > >
> > > On RL for state tracking:  yes, our results suggest it is a promising direction. CogniLoad’s verifiable metadata makes it feasible to reward the behaviors that matter.
> > >
> > > Practically this could mean:
> > > - Using step-wise verifiable rewards for correct state transitions plus a sequence-level consistency reward and penalizing spurious or missed updates and instruction non-compliance.
> > > - Training with a curriculum over (d, N, ρ), with targeted stress on the 25–50 % distractor band where models are most brittle.
> > > - Encourage explicit intermediate state representations (e.g., constrained “state tables”) and self-checks as part of the policy.
> > > - Diversify attribute ontologies (which can be changed trivially in the provided generation code) to avoid overfitting to templates to evaluate out-of-distribution generalization.
> > > - Report gains via ECL50/ID50/NT50 to show whether improvements come from better state tracking vs just better formatting
> > >
> > > Complementary to RL, architectural changes (e.g. memory-efficient attention, KV cache compression, retrieval or sliding-window schemes, verifier-assisted decoding) address the same failure modes and can be combined with the above. The three independent parameters of CogniLoad within one task offer efficient diagnostics on the impact of these targeted architecture ablations beyond existing benchmarks.

---

> > ### Comment · Reviewer_aNzL · 2025-11-26
> >
> > Thanks for your response!
> >
> > To clarify - I meant a theoretical verification of the possibility of the problems. Many of the problems in this benchmark require a large number of tokens to fully state. If the model was to write out the full state at each step, would all of the problems fit within the context window?
> >
> > Sorry that I couldn't find the code link. Thank you for sharing it.
> >
> > In thinking about difficulty - math competitions such as AIME are considered less difficult than the IMO, but this doesn't seem like a difference in volume, but in kind. How does this fit within the framework you describe, and how does d capture this type of difficulty?

---

> ### Author Response · Authors · 2025-11-26
>
> Thank you for your response and interest in our work!
>
> ## Theoretical verification
> To verify this, let's consider the most extreme parameter configuration of $N=250$ statements with Intrinsic Difficulty $d=10$.
>
>  Based on the data from Appendix H we know the prompt takes $\approx 16'500$ tokens for these problems. This means around $32'768$ - $16'500$ =  $16'268$ tokens remain for the states with our context window of 32,768 tokens.
>
> To solve the puzzle the LLM only needs to track the state of the Person of Interest (PoI) which has 10 attributes (at $d=10$) while stating one attribute of the state requires approximately $\approx$ $4\text{-}6$ tokens (e.g. blue jeans, kitchen, red socks, pink hat, ...). This means the LLM needs around $\approx 40\text{-}60$ tokens per step.
>
> For 250 steps (at $N=250$) this results in $\approx 10'000\text{-}15'000$ tokens which is lower than the $\approx 16'268$ tokens remaining.  So yes, all the problems would fit into the context window if the LLM just wrote out the state of the PoI each step.
>
> The estimate of $\approx 4\text{-}6$ tokens per attribute is conservative and consistent with the output tokens we report in Appendix H empirically for the $N=250,d=10$ configuration (e.g. GPT-5 needing on average $9'283$ output tokens with the 95th percentile being at $12'365$ output tokens).
>
> ## How does d capture the type of difficulty of AIME vs IMO ?
>
> Thank you for the AIME vs IMO analogy because it maps directly to the distinction we are trying to capture between *volume* and *kind* of difficulty. In our framework, *volume* is controlled by the task-length parameter $N$ (how many sequential updates must be processed), whereas *kind* is controlled by the intrinsic-difficulty parameter $d$, which changes the structure of each reasoning step.
>
> Concretely, increasing $d$ does four things at once in every puzzle:
> - it introduces more people and more attribute categories,
> - it enlarges the set of possible values for each attribute,
> - it allows each statement to *condition* on more attributes of a person simultaneously, and
> - it allows each statement to *update* more attributes at once.
>
> At low $d$, rules are essentially “if A then B” for a single attribute. At high $d$, a single rule can say: “if a person simultaneously has properties A, B, C, …, then update several of their properties.” This forces the model to track and apply multi-attribute, many-way dependencies at every step, which is exactly what Cognitive Load Theory in its definition of Intrinsic Cognitive Load describes as increased “element interactivity” [1].
>
> The difficulty therefore comes from higher relational complexity and deeper logical inter-dependencies, not from more steps (i.e., volume). In that sense, higher $d$ is intended to capture the same qualitative “difference in kind” you point to in moving from AIME‑style to harder IMO‑style problems (within a fixed puzzle family).
>
> [1] Halford, Graeme S., William H. Wilson, and Steven Phillips. "Processing capacity defined by relational complexity: Implications for comparative, developmental, and cognitive psychology." Behavioral and brain sciences 21.6 (1998): 803-831.

---

### Official Review · Reviewer_kvXx · 2025-10-29

**Soundness:** 3
**Presentation:** 3
**Contribution:** 3
**Rating:** 6
**Confidence:** 3

**Summary:**

CogniLoad introduces a synthetic benchmark grounded in Cognitive Load Theory (CLT) to evaluate LLMs' long-context reasoning capabilities. This benchmark independently controls three dimensions: intrinsic difficulty, task length, and distractor density.

Key contributions include: (1) operationalizing CLT for LLM evaluation, (2) revealing task length as the dominant performance constraint, (3) identifying U-shaped sensitivity to distractors, and (4) providing capacity thresholds (e.g., ECL50, ID50) for 22 models.

**Strengths:**

**Originality​**

Introduces the first benchmark grounded in Cognitive Load Theory (CLT) for LLMs, disentangling intrinsic difficulty (d), distractor density (ρ), and task length (N) to diagnose reasoning failures.

​**Quality​**

Rigorous synthetic puzzle generation with independent parameter control, validated through large-scale evaluation and GLM-based capacity thresholds (ECL50, ID50).

​**Clarity​**

Well-structured with explicit CLT-LLM mapping. The operationalization of task length (N) as a proxy for germane load (GCL) could be further justified; visualizations (e.g., U-shaped ρ sensitivity) enhance interpretability.

​**Significance​**

Sets a new standard for interpretable LLM evaluation, revealing task length (N) as the dominant bottleneck and providing actionable "cognitive fingerprints" for model improvement.

**Weaknesses:**

**Limited Generalizability Beyond Deductive Reasoning**

The benchmark exclusively tests ​deductive logic puzzles, neglecting inductive/abductive reasoning or real-world tasks (e.g., math, code debugging). This restricts insights into broader reasoning capabilities.

**Questions:**

1. The factorial design isolates each dimension, but are there ​interaction effects​ (e.g., high d+ high ρexacerbating errors)?

---

> ### Author Response · Authors · 2025-11-18
>
> ## Limited Generalizability Beyond Deductive Reasoning
>
> We acknowledge this trade off in our limitations (see “Scope of Reasoning and Generalizability” in Discussion section) as it is necessitated by the factorial design and the synthetic generation of our benchmark that makes it data contamination proof (by enabling the generation of randomized new tests on the fly). Evaluating also inductive and abductive reasoning styles, would lead to different task semantics (that invalidate direct comparisons) each requiring a different operationalization of CLT factors and dedicated new generation algorithms. Alternative recent benchmarks like LogiEval (which we will cite in the revised paper), test all three types of reasoning but forgo all the benefits of CogniLoad (e.g. variation of independent dimensions, single task semantics, synthetic generation, ...).
>
> ## The factorial design isolates each dimension, but are there interaction effects (e.g., high d + high $\rho$ exacerbating errors)?
>
> We conducted a thorough analysis of all interaction effects that we report and discuss in section B of the updated appendix. Please see also our response to reviewer xtvk where we discuss these interaction effects in the context of internal LLM mechanisms.

---

> ### Author Response · Authors · 2025-11-27
>
> Dear Reviewer kvXx,
>
> Thanks again for your efforts and interest in our paper. As the deadline for the author-reviewer discussion phase is nearing, we would like to ask if our responses have effectively addressed the concerns you raised. Should any remain unresolved, please let us know, and we will promptly follow up.
>
> If we have sufficiently addressed your concerns, we hope you could re-evaluate our work more positively.
>
> Thank you for your time and help.
>
> Best regards,
> CogniLoad authors

---

### Official Review · Reviewer_xtvk · 2025-10-29

**Soundness:** 2
**Presentation:** 3
**Contribution:** 2
**Rating:** 6
**Confidence:** 2

**Summary:**

The paper introduces CogniLoad, a synthetic long‑context reasoning benchmark grounded in Cognitive Load Theory (CLT). It factorizes task stressors into three independently tunable dimensions: (1) intrinsic difficulty d (element interactivity and rule complexity), (2) distractor density ρ (needle‑to‑hay ratio), and (3) task length N (as an operational proxy for sustained “germane‑like” processing). The benchmark generates natural‑language logic‑grid puzzles with sequential state updates, evaluates exact‑match accuracy, and provides an error taxonomy. They have run factorial evaluations for 22 LLMs and fit a per‑model binomial GLM to derive interpretable “capacity thresholds” (ECL50, NT50, ID50). The main empirical findings are that length N is the dominant constraint, that distractors exhibit a characteristic U‑shaped performance curve with worst accuracy at intermediate ρ, and that models differ meaningfully in their sensitivity profiles.

**Strengths:**

The design isolates three often‑confounded aspects of long‑context reasoning and does so in a way that scales to arbitrarily long sequences. The generator is cleanly specified with validation constraints and produces semantically coherent, non‑trivial distractors. The regression view and derived thresholds offer compact, model‑specific “cognitive fingerprints” that go beyond aggregate accuracy; this is a useful diagnostic interface for both research and engineering. The error taxonomy and supplementary analyses (e.g., AIC comparison for quadratic ρ, confidence intervals) increase interpretability and make the empirical claims credible. The reproducibility is strong: the code repository and dataset release enable immediate verification.

**Weaknesses:**

The CLT analogy is motivating but only partially validated. Using sequence length N as a proxy for germane load conflates at least three mechanisms in LLMs - attention budget, recurrent state tracking over updates, and compounding interference from earlier updates. The paper acknowledges this, but the empirical analysis does not probe which of these mechanisms dominates or whether N’s effect interacts mechanistically with d and ρ in predictable ways.

The evaluation mixes performance and infrastructure limits. The max‑context failures are treated as an error category, but 32K token limits, tokenization differences, prompt overhead, and vendor defaults create avoidable confounds between “reasoning under load” and “serving limits.” Several models show many “long‑context budget overflow” errors at N=250; this is partly a systems artifact rather than a cognitive limitation. The paper would be stronger if it normalized prompt and decoding budgets across models, used a shared short instruction template, and pre‑tested that all N regimes fit for at least a subset of items per model.

Evaluation relies on an exact match with a final one‑sentence answer. The authors mitigate formatting drift via post‑processing and categorize “last‑logic” failures, but the metric remains coarse. Without step‑wise verification, it is hard to separate state‑tracking mistakes from late‑stage overwrites or local misparses. Given that the generator has full ground‑truth state trajectories, a step‑fidelity metric or a probabilistic grading of intermediate updates would add substantial value and make the “germane‑like” interpretation more concrete.

**Questions:**

1. Please report, in the main paper, at least a compact interaction analysis among d, N, and ρ (one table with key coefficients or ANOVA‑style significance and effect sizes). The appendix suggests clear qualitative patterns; quantifying them would greatly help interpretation.
2. How sensitive are the main conclusions to the instruction format? If the final answer is allowed to be free‑form (no strict one‑sentence requirement), do “last‑logic” errors diminish while overall accuracy remains similar?
3. What is the average token length of puzzles at each N and d? Reporting token distributions per condition and per model tokenizer would help separate system limits from reasoning load.
4. Can you provide size‑matched comparisons with harmonized decoding settings (temperature, top‑p, max tokens) across representative models, to reduce variance induced by vendor defaults?

---

> ### Author Response · Authors · 2025-11-18
>
> ## The CLT analogy is motivating but only partially validated.
>
> To probe whether and how these mechanisms co-vary with intrinsic difficulty $d$ and the needle ratio $\rho$, we added an explicit interaction analysis to the updated Appendix (see section B). The new results show that $N$’s effect is not a uniform “length penalty,” but is contingent on $d$ and $\rho$ in predictable ways:
>
> -  Strong joint effect of difficulty and length. A robust, negative $d\times N$ appears in 17/22 models (likelihood–ratio $p_{\mathrm{LR}}<0,01$), indicating super-additive degradation when tasks are both longer and intrinsically more complex - consistent with compounding state-tracking and interference, beyond a separable attention-budget effect of $N$.
> -  Difficulty amplifies distractor harm. A negative $d\times \rho$ is significant in 13/22 models ($p_{\mathrm{LR}}\le 0,01$), i.e., higher intrinsic complexity makes models more vulnerable to extraneous load. This aligns with the idea that more complex schemas are more fragile to distractors and frequent updates to the state of the Person of Interest (PoI).
> -  Length–distractor interplay differentiates mechanisms. The $N\times \rho$ interaction is significant in a subset of models and its sign separates two regimes:
>     - Positive $N\times \rho$ (e.g., Qwen3‑30B‑A3B, EXAONE‑Deep‑32B) suggests that increasing $\rho$ mitigates the length penalty. This is consistent with “distractor reduction” dominating at longer sequences (attention/retrieval bottleneck).
>     - Negative $N\times \rho$ (e.g., DS‑Qwen‑7B; gemini‑2.5‑pro, which also shows long-context overflows) indicates that longer sequences exacerbate the harm of more frequent PoI state changes. This is consistent with recurrent state-tracking and interference dominating.
> -  Three-way moderation. A positive $d\times N\times \rho$ is significant in 12/22 models, indicating that when tasks are simultaneously long and complex, increasing the share of needles has a larger beneficial marginal effect (i.e., reducing extraneous load particularly helps when state complexity and update depth are both high). This is consistent with the intuition that reducing distractors yields the greatest benefit when both intrinsic complexity and sequence length are large, even though $\rho$ also increases the number of PoI updates that must be tracked.
>
> While we cannot uniquely isolate the three mechanisms without targeted ablations, we hope these interaction patterns address your concern: the impact of $N$ is mechanistically contingent on $d$ and $\rho$ across model families and capacity tiers. Many models show super-additive difficulty–length coupling, difficulty-amplified distractor harm, and model-dependent length–distractor trade-offs that match the hypothesized attention, state-tracking, and interference components.
>
> ## The evaluation mixes performance and infrastructure limits
>
> In very few open weights models (i.e. the Phi-4 family) we observe that in the transparent reasoning traces that these errors primarily occur due to indefinitely recurring reasoning loops and models “going crazy” by repeating the same tokens indefinitely - here it is not an infrastructure limit but a genuine failure to solve the task.
>
> Meanwhile, in the Gemini 2.5 family of models, we notice a generally higher token usage also for N<250 problems which is the likely cause of these max-context issues at N=250. You are correct that this reflects an infrastructure limit rather than a performance concern.
>
> To ensure fairness, since this is not a benchmark about reasoning efficiency, we are re-evaluating the Gemini 2.5 family with increased token limits for N=250 problems to decouple reasoning capability from verbosity constraints. We are currently re-running these experiments and will update the manuscript as soon as the results are available.
>
> However, we would like to re-emphasize that this is not a systematic problem (i.e. N=250 problems are generally solvable within the 32K token limit) because the max-context errors for the vast majority of models is negligible and models succeed and fail with logic-errors within this context limit. The analysis of the output length distribution included in the updated Appendix confirms this.

---

> ### Author Response · Authors · 2025-11-18
>
> ## Evaluation relies on an exact match with a final one-sentence answer.
>
> The primary reason we opted for this coarse metric was that it allows the evaluation of both open and proprietary frontier models, which do not expose the reasoning traces.
>
> While the metadata for each puzzle enables a more fine grained analysis of the failure type, the different styles of free form reasoning in the traces and phenomena we observed in them (e.g. backtracking, recursive reasoning loops, ..)  would necessitate complex and robust deterministic heuristics, probabilistic methods, or a separate “LLM-as-judge” to trace each intermediate reasoning steps. Each one of these approaches may introduce idiosyncratic errors while complicating the programmatic evaluation and the interpretation of its results.
>
> Being primarily concerned about the question “Did the reasoning LLM solve the problem correctly?”, and considering the scale of the benchmark (14k puzzles per model), we believe evaluating the task success as an exact match provides a deterministic and reproducible signal that correlates strongly with the reasoning success that we seek to measure.
>
> ## Question: Please report, in the main paper, at least a compact interaction analysis among d, N, and p
> As noted in our response to concern 1, we have conducted this interaction analysis and updated the appendix to add a section containing this analysis, its quantitative results, and their discussion.
>
> ## Question: How sensitive are the main conclusions to the instruction format?
>
> In the evaluation, we also accept alternative formatting of the final sentence to mitigate this sensitivity in the results.
>
> We ultimately count as correct the outcomes:
> - correct_poi: e.g. “Peter’s final state is Yellow” in the last sentence mentioning “Peter” (the PoI) by name
> - correct_last_sentence: e.g. “Final Answer: yellow” as the last sentence of the response.
>
> While we mention these specific outcomes in the Appendix (“C.5 Buckets, precedence, and accuracy”), we recognize that this feature of the evaluation pipeline should be mentioned more prominently and we will do so in the updated paper. We omitted the count of correct_poi and correct_last_sentence cases per model, d and N in the appendix due to the horizontal page limit and because they measure roughly the same format drift that poi-logic and last-logic already indicate.

---

> ### Author Response · Authors · 2025-11-18
>
> ## Question: What is the average token length of puzzles at each N and d?
>
> We have updated the appendix with section G that reports the average token and 90th percentile lengths (prompt and output) per model at different N and d. The tables in the appendix provides the requested quantitative characterization per condition and per tokenizer: prompt tokens quantify the input length at each (N, d) with minimal tokenizer variance, while output tokens reveal model‑specific verbosity and heavy tails at large N. The token-length distributions also address your previous concern about confounding system limits with reasoning load.
>
> At fixed (N, d), prompt lengths are nearly identical across tokenizers, showing that the presented input burden is the same for all models in a given condition. The substantial differences arise in output tokens: as N grows, some tokenizers exhibit heavy‑tailed output distributions that drive total context use to the 32 768‑token ceiling. This is most apparent for the Gemini 2.5 models at N = 250, where output p90 values often lie in the mid 20 000s and push the combined prompt + output near or beyond the budget. By contrast, o3 and the GPT‑5 family (and many mid‑tier open models) keep outputs much shorter under the same prompts, leaving comfortable headroom indicating their degradation at long N is therefore better attributed to reasoning under load (state tracking, logic) than to hard serving limits.
>
> As mentioned and consistent with this finding, we will re‑run the high‑N regimes for the Gemini 2.5 models with a higher max‑context, so their results do not reflect a systems artifact. For the majority of other models, total tokens already sit comfortably below the 32 K budget under the same (N, d) conditions. We are currently re-running these evaluations currently and will update the manuscript as soon as results are available.
>
> ## Question: Can you provide size-matched comparisons with harmonized decoding settings (temperature, top-p, max tokens) across representative models, to reduce variance induced by vendor Defaults?
>
> We have not studied variations in the decoding settings, because usually when models are evaluated on benchmarks or used by users, they are using these vendor defaults for decoding. We believe it’s difficult to harmonize them (i.e. temperature, top-p) because it is not clear how these settings are influenced by the LLMs being trained under different regimes and with different underlying architectures. The existence of these diverging defaults indicates that each LLM by default produces logits at different scales, and therefore the settings are highly model specific. We therefore expect model vendors to suggest the “best” general inference settings for these defaults, and notably the GPT-5 series of models doesn’t even allow developers to set the temperature and top-p anymore. Furthermore, tuning these hyperparameters for each model risks overfitting the benchmark to specific settings, while vendor defaults represent the intended 'general purpose' capability of the model.
>
> The max-tokens setting is a special case, and as we state in our response to concern 2, we will increase it to reevaluate Gemini 2.5 with a setting that makes all the N=250 are solvable in light of the model's generally higher token verbosity.

---

> ### Author Response · Authors · 2025-11-27
>
> Dear Reviewer xtvk,
>
> Thanks again for your efforts and interest in our paper. As the deadline for the author-reviewer discussion phase is nearing, we would like to ask if our responses have effectively addressed the concerns you raised. Should any remain unresolved, please let us know, and we will promptly follow up.
>
> If we have sufficiently addressed your concerns, we hope you could re-evaluate our work more positively.
>
> Thank you for your time and help.
>
> Best regards,
> CogniLoad authors

---

### Official Review · Reviewer_je6v · 2025-10-31

**Soundness:** 2
**Presentation:** 3
**Contribution:** 3
**Rating:** 6
**Confidence:** 4

**Summary:**

This paper proposes a new benchmark, CogniLoad, that leverages concepts from the Cognitive Load Theory to design a logic grid puzzle reasoning benchmark that systematically evaluates LLMs on their *intrinsic* load (ICL), *extrinsic* load (ECL), and *germane* cognitive load (CGL), with the last one being approximated via task length $N$. The authors evaluate 22 SOTA LLMs and present several trends and insights on the scaling of each of these dimensions of reasoning ability, for different classes of LLMs.

**Strengths:**

* CogniLoad is a systematic benchmark and the first to make proper distinction between the 3 studied kinds of cognitive load LLMs have to deal with when solving hard reasoning problems.

* The work is grounded in a formalism from cognitive science, namely Cognitive Load Theory. Although some aspects (in particular, the connection to germane load) is a bit of a stretch, the notions of intrinsic and extrinsic load align very well with the notions of actual task and distracting information.

* The study is quite detailed, with many LLMs of various kinds evaluated on the proposed benchmark. The authors discuss several trends. While some of these (e.g., increasing the intrinsic difficulty reduces LLM accuracy) are both intuitive and, to some extent, reported before, the present paper does it carefully while keep all (or most) other variables constant. This, thus, provides a cleaner evidence. Additional analyses, like failure modes and differential sensitivity, are useful.

* The paper was easy to read and understand.

**Weaknesses:**

* The paper can use a clearer description of the *germane* load from the CLT theory -- what it means, how it manifests, etc. -- along with some more discussion of why the task length parameter, $N$, is a good proxy for it.

* (stronger concern) The needle-to-hay parameter, $\rho$, is not as clean cut as intended, and this might even explain the U-shaped performance the authors observe. Specifically, the $\rho$ parameter mixes two distinct aspects: higher $\rho$ means more statements about the PoI (person of interest), which has two competing effects: (a) it means fewer distractor statements, which makes the problem easier to solve; and (b) it means a higher number of state changes for PoI, which makes the problem harder to solve. As an example, as one moves from $\rho = 5\%$ to $\rho = 10\%$, the number of distractor statements drop by (only) $5\%$, but on the other hand, the number of state transitions for PoI go up by 2x, i.e., double!! The authors don't discuss this important aspect and its implications. Note that this provides a reasonable explanation of the U shaped curve observed -- at very low values of $\rho$, the number of state transitions dominate the accuracy (the relative change in distractors isn't much), while at very high values of $\rho$, the relative change in the number of distractors is the one that's significant. I would ask the authors to look more deeply into this and see how it affects the message of and explanations in the paper.

* While it's good to have a formal puzzle definition in section 2, it is also important to verbally convey what each of the constraints is achieving. Can you add some explanatory text?

* For task length $N$, and especially at relatively higher $\rho$ values (which together means many state transitions for PoI), there is good reason to expect models to struggle without chain-of-thought reasoning, i.e., "non-reasoning models", based on theoretical results showing that transformers (without CoT) are inherently parallel (in complexity class $TC^0$ [1,2]) while state-tracking is a sequential task (in general well-known to be complete for the stronger complexity class $NC^1$). It's also known formally that using CoT allows models to perform state tracking and other sequential tasks [3,4]. This might explain some of the observed behavior, e.g., GPT-5 and o3 being resiliant to increase in $N$. Including some discussion of this connection would be valuable.

[1] The Parallelism Tradeoff: Limitations of Log-Precision Transformers, Merrill et al., 2022

[2] Transformers in Uniform TC0, Chiang, 2024

[3] The Expressive Power of Transformers with Chain of Thought, Merrill et al., 2024

[4] Chain of Thought Empowers Transformers to Solve Inherently Serial Problems, Li et al., 2024

**Questions:**

* Any thoughts on the $\rho$ parameter not quite being as clean as anticipated, as increasing it makes the problem both easier and harder in different ways? (see note above)

---

> ### Author Response · Authors · 2025-11-16
>
> ## Clearer description of the germane load from the CLT theory and why the task length parameter, N, is a good proxy for it.
>
> Thank you for bringing the short description of germane load (GCL) to our attention. In Cognitive Load Theory, GCL denotes the proportion of working‑memory resources productively allocated to constructing and maintaining task‑specific schemas [1,2]. Because GCL reflects how cognitive resources are allocated by the learner rather than a property of the task itself (unlike intrinsic and extraneous load), we need to operationalize conditions that elicit germane processing via proxy.
>
> In CogniLoad, the schema to be maintained by the learner (i.e., the LLM) is the Person‑of‑Interest’s attributes across \(d\) categories, and each statement requires a sequential schema application that updates this schema. Critically, \(N\) increases the number and duration of these schema applications without increasing intrinsic load (per‑step element interactivity and state dimensionality) or extraneous load (distractor density). This is why we use \(N\) as an operational proxy for germane‑like processing (i.e. sustained, constructive engagement with the task‑relevant schema). In simpler non-CLT terminology, increasing N scales the number of times the LLM needs to update the state of the PoI and this update action is what requires “cognitive resources” for the LLM. We will make this clearer in the manuscript.
>
> A prominent problem setting in CLT noted for high germane load is the “worked‑example” format, in which successive problem states are presented together with the next move (i.e. the learner “merely has to look at each problem state and the move leading to the next state” [2]). This closely mirrors a sequence step in CogniLoad and increasing \(N\) scales the amount of steps and respectively the schema maintenance/applications required for solving the whole puzzle, rather than the intrinsic interactions per step.
>
> [1] Paas, Fred, Alexander Renkl, and John Sweller. "Cognitive load theory and instructional design: Recent developments." Educational psychologist 38.1 (2003): 1-4.
>
> [2] Sweller, J. (2010). Element interactivity and intrinsic, extraneous, and germane cognitive load. Educational Psychology Review, 22(2), 123-138. doi:https://doi.org/10.1007/s10648-010-9128-5
>
> ## Two effects at play in $\rho$
>
> We agree there are these two competing effects.
>
> Indeed, it is surprising that for most models a problem with $\rho$=5% (i.e. few necessary deductions, high noise) is roughly as difficult as a problem with $\rho$=95% (i.e. mostly necessary deductions, low noise) and similarly for $\rho$=25% vs $\rho$=75% (or $\rho$=10% vs $\rho$=90%) due to the usually symmetric shape of the curve. Interestingly, the sign and magnitude of [Acc(low $\rho$) - Acc(high $\rho$)]  (with Acc standing for Accuracy) indicates which of these two effects relatively dominates in a model:
>
> - If  [Acc(low $\rho$) - Acc(high $\rho$)] ~ 0 (eg DeepSeek-R1, GPT-5 mini, …): The two effects balance each other. A problem with high noise has the same performance as a problem with a high number of essential statements.
> - If  [Acc(low $\rho$) - Acc(high $\rho$)] > 0 (e.g. GPT-5, o3, EXAONE, ..): The model is less distracted by noise. It benefits from the few essential statements in the high noise tasks and this benefit outweighs the performance penalty of the added noise.
> - If  [Acc(low $\rho$) - Acc(high $\rho$)] < 0 (e.g. Gemini-2.5-flash, DeepSeek-R1-LLama, ..): The model is more affected by noise. Adding noise to the few essential statements outweighs the benefits that this relatively easier puzzle has.
>
> To statistically substantiate this observation, we conducted an interaction analysis among \(d\), \($N$\), and \($\rho$\) (see updated Appendix section B for details). The signs and significance of \(N$\times \rho\$), \(d$\times \rho$\), and \(d$\times N\times \rho$\) diagnose which side of the trade-off dominates for each model and regime: positive \(N$\times \rho$\) indicates that reducing distractors (higher \($\rho$\)) mitigates length-related degradation, negative \($N\times \rho$\) or negative \($d\times \rho$\) indicate that frequent PoI state changes (higher \($\rho$\)) exacerbate tracking/interference costs (especially at higher \(N\) or \(d\)), and a positive three-way term shows that increasing \($\rho$\) is particularly helpful when tasks are both long and complex.
>
> This analysis reconciles the observed U-shape and its asymmetries without additional assumptions, and we integrate these results into the paper.

---

> > ### Author Response · Authors · 2025-11-16
> >
> > ## Verbal explanation of constraints
> >
> > Agreed. We will add the following text in §2.1.4:
> > - Hay cannot affect the PoI and cannot collapse all non-PoIs onto the PoI: Ensures a hay statement is truly extraneous and that further hay statements can be generated through at least one non-PoI having a different state than the PoI.
> > - Needles cannot affect everyone and cannot collapse all non-PoIs to the PoI: Prevents the diversity of state attributes from collapsing while avoiding steps that would make the non-PoIs become identical to the PoI and thereby prevent the generation of further genuine hay statements.
> > - Hay statements cannot make all non-PoIs become identical: Prevents the state of all non-PoIs essentially collapsing into a single non-PoI state and is particularly important at low difficulties where this would trivialize further statements and their solution.
> >
> > We will also add one sentence in §2.1.2 that the initialization ensures people differ in all attributes, guaranteeing a maximally diverse starting point.
> >
> > ## Complexity class and CoT influencing state tracking
> > Thank you for bringing these formal and theoretical results to our attention. They are highly relevant and we will discuss them in the context of these complexity classes in the revised paper. They are also consistent with our initial experiments where we tried non-reasoning LLMs on CogniLoad and observed that they fail catastrophically already at very low N = 5 or 10. This motivated our decision to exclusively evaluate reasoning models in the paper.
> >
> > ## Thoughts on $\rho$ not being "clean"
> > We respond to this question in depth in our response to "Two effects at play in $\rho$". We wouldn't characterize $\rho$ as "unclean" but as a parameter that allows a more differentiated insight on the impact of noise. To the best of our knowledge, no previous benchmark evaluating distractions allowed for independent control of N and relative noise $\rho$, so we believe it is an important contribution to unmask this tradeoff between the two mentioned effects.

---

> ### Author Response · Authors · 2025-11-27
>
> Dear Reviewer je6v,
>
> Thanks again for your efforts and interest in our paper. As the deadline for the author-reviewer discussion phase is nearing, we would like to ask if our responses have effectively addressed the concerns you raised. Should any remain unresolved, please let us know, and we will promptly follow up.
>
> If we have sufficiently addressed your concerns, we hope you could re-evaluate our work more positively.
>
> Thank you for your time and help.
>
> Best regards,
> CogniLoad authors

---

### Meta-Review · Area_Chair_JxZw · 2026-01-07

**Summary:**

This paper presents *CogniLoad*, a synthetic long-context reasoning benchmark inspired by Cognitive Load Theory, with three independently controlled dimensions: intrinsic difficulty $d$, distractor density $\rho$, and task length $N$. Reviewers generally found the benchmark carefully engineered and the empirical study thorough, but also raised several conceptual and methodological concerns. In particular, there was hesitation about how faithfully CLT notions—especially germane load—map onto LLM behavior, whether $\rho$ conflates multiple opposing effects, and whether some observed failures reflect reasoning limits versus system or prompting artifacts (je6v, xtvk, aNzL). Overall scores clustered around weak accept. After the rebuttal, the paper feels borderline but leaning slightly positive: the authors made an effort to address the critiques, yet some high-level concerns remain only partially resolved.

**Reviewer Concerns:**

Several technical issues were reasonably addressed in the rebuttal. The authors provided a clearer justification for using $N$ as a proxy for germane-like processing and added interaction analyses among $d$, $N$, and $\rho$, which directly speak to concerns from je6v and xtvk. The discussion of the U-shaped effect in $\rho$ and the acknowledgment of two competing mechanisms help clarify, but do not entirely eliminate, the feeling that this parameter mixes “easier” and “harder” effects in a way that complicates interpretation (je6v). Likewise, token-length analyses and planned re-runs reduce, but do not fully erase, concerns about context-window confounds for some models (xtvk, aNzL). More broadly, questions about the benchmark’s scope—limited to deductive, state-tracking style reasoning—and about whether the insights go substantially beyond a cleaner re-packaging of known weaknesses remain open (kvXx, aNzL). These are not fatal flaws, but they do justify the borderline nature of the submission.

**Reviewer Scores:**

All reviewers initially scored the paper around 6, and none re-engaged after the author responses. Although the authors responded in detail, the rebuttal primarily adds interpretation, interaction analyses, and planned adjustments rather than substantially new empirical evidence or new experimental results. As a result, I would not expect large score changes. Reviewers such as je6v and xtvk might have slightly increased confidence that their concerns were taken seriously, but likely without changing the numerical score. Reviewers kvXx and aNzL, who already leaned weak accept, would probably remain in that position. Overall, the paper remains a borderline case, with modestly improved clarity but largely unchanged reviewer sentiment.

---

### Decision · Program_Chairs · 2026-01-26

Accept (Poster)